# Cortical responses to balance perturbations persist without active postural control

Daphne N. R. Jansen [ID], Lucas Mensink [ID], Matto Leeuwis [ID] and Patrick A. Forbes [ID]

*Department of Neuroscience, Erasmus MC, University Medical Center Rotterdam, Rotterdam, The Netherlands*

Handling Editors: Richard Carson & Bettina Schwab

The peer review history is available in the Supporting Information section of this article (https://doi.org/10.1113/JP290280#support-information-section).

**Abstract figure legend** A robotic balance simulator was used to test whether cortical responses to balance perturbations are contingent on active balance control or instead reflect the detection of unexpected motion. Participants experienced

**Daphne N. R. Jansen** completed a Bachelor in Biology at Leiden University, a Master in Neuroscience at Erasmus University Rotterdam and a Master in Philosophy at Vrije University Amsterdam. During her Master in Neuroscience, she did her research internship at Patrick Forbes' Sensorimotor Neuroscience and Biorobotics Lab, supervised by Patrick Forbes and Lucas Mensink. Her project studied cortical signals underlying balance control, and inspired her to pursue research on the functional roles of neural signals and the relationship between the body, brain and mind. **Lucas Mensink** began his PhD in 2022 after completing a Bachelor in Psychobiology at the University of Amsterdam and a Master in Human Movement Sciences at Vrije University Amsterdam. He studies how the brain stabilizes our inherently unstable posture using robotic simulation and neurophysiology under the supervision of Patrick Forbes at the Erasmus MC. His research focuses on how noise and uncertainty shape balance control, perception and their underlying neural signals.

D. N. R. Jansen and L. Mensink contributed equally to this work.

The Journal of Physiology

identical support-surface rotations (toes-up and toes-down) while actively controlling balance or while being moved passively without control. Removing control strongly reduced balance-correcting EMG responses (up to 60%), yet the midfrontal balance N1 potential (occurring at 100–200 ms) decreased only modestly (∼10%). This dissociation indicates that early cortical responses to balance perturbations persist when active postural control is removed and probably reflect the detection and evaluation of unexpected sensory events.

**Abstract**  Standing balance relies on rapid reflexes as well as longer-latency subcortical and cortical processes to generate corrective responses to postural disturbances. EEG studies consistently identify two perturbation-evoked markers of cortical activity, the balance N1 and midfrontal theta power, associated with changes in body orientation and corrective actions. It remains unclear, however, whether these markers depend on the nervous system's active control of posture or reflect a more general evaluation of unexpected sensory input. We tested this by measuring cortical and muscle activity during support-surface perturbations while systematically manipulating whether or not participants actively controlled posture. In Experiment 1 ($n = 10$), participants experienced identical perturbations while either actively balancing or being passively moved through equivalent motion. Despite large reductions in balance-correcting muscle activity during passive trials (∼30–60%), N1 and theta responses persisted with only modest amplitude reductions (∼10%). In Experiment 2 ($n = 16$), we created passive conditions increasingly removed from balance by varying sensory feedback (footplate + whole-body *vs.* footplate-only motion) and motor engagement (isometric contraction *vs.* relaxed posture). Relaxed postures markedly suppressed muscle responses, yet cortical responses persisted, showing only modest modulation with sensory feedback (larger during footplate-only rotations) and no dependence on motor engagement. Together, these results indicate that N1 and midfrontal theta are not dependent on active postural control and persist even without matching sensory feedback or motor engagement. Rather than reflecting the generation or scaling of corrective actions, they index the early detection and evaluation of unexpected sensory events, consistent with prediction error or surprise processing.

(Received 7 October 2025; accepted after revision 23 January 2026; first published online 23 February 2026)

**Corresponding author** P. A. Forbes: Department of Neuroscience, Erasmus MC, University Medical Centre Rotterdam, Rotterdam, The Netherlands.    Email: p.forbes@erasmusmc.nl

**Key points**

- When standing balance is disturbed by a perturbation, the brain shows characteristic electrical responses called the balance N1 and theta activity, which are thought to contribute to balance-correcting actions.
- We tested whether these cortical responses depend on actively controlling posture or instead reflect the detection of unexpected motion irrespective of balance conditions.
- Participants stood in a robotic balance simulator and experienced identical perturbations while actively balancing or being passively moved, and when whole-body sensory feedback and muscle engagement were removed.
- The balance N1 and theta activity persisted in conditions where participants were not controlling their movement and even when whole-body sensory feedback and motor engagement were removed, whereas balance-correcting muscle responses were strongly diminished.
- This shows that cortical responses to balance perturbations are not specific to active balance control but probably represent the brain's detection and evaluation of unexpected sensory events.

## Introduction

Humans regulate standing balance by sensing and adjusting the body's orientation and movement in space.

This process relies on neural circuits that integrate sensory signals from the vestibular, somatosensory and visual systems to generate effective postural corrections in response to perturbations that threaten balance (Forbes

et al., 2018; Prochazka, 1989; Rasman et al., 2018). Although reflexive and subcortical mechanisms are primarily responsible for corrective balance responses, cortical structures are thought to govern the refinement and adaptation of these behaviours when needed (Bolton, 2015; Jacobs & Horak, 2007). Accordingly, recent neuro-imaging studies have focused on identifying the cortical contributions to balance control, particularly during externally applied perturbations. For instance, EEG has been used to measure brain activity during balance disturbances, revealing key markers – the balance N1 and theta activity – that may reflect cortical involvement in reactive balance control (Bolton, 2015; Payne, Ting, et al., 2019). The balance N1 is a negative component of an event-related potential (ERP), peaking approximately 100–200 ms after the onset of a postural disturbance (Bolton, 2015). Theta activity, defined as oscillations in neural activity between 4 and 7 Hz, emerges over a similar time window (Peterson & Ferris, 2018; Stokkermans et al., 2022). Both markers have been localized to the supplementary motor area (SMA; Marlin et al., 2014; Peterson & Ferris, 2018), a brain region involved in motor planning and execution. This has often been taken as evidence for a role in balance control, yet whether these cortical responses are truly specific to postural control or instead reflect more general sensorimotor processing remains uncertain.

Several balance-related factors have been found to influence both cortical activity and reactive balance responses, including the instructed motor task (Quant et al., 2004; Weerdesteyn et al., 2008), postural threat (Adkin et al., 2008; Stokkermans et al., 2022) and repeated exposure to perturbations (Maki & Whitelaw, 1993; Mierau et al., 2015; Payne, Hajcak, et al., 2019; Welch & Ting, 2014). These findings have motivated attempts to link N1 and theta activity to the corrective motor responses necessary to maintain posture. For example, larger N1 amplitudes were observed during stepping responses compared to feet-in-place adjustments, a difference interpreted as evidence of a relationship between cortical activity and corrective motor output (Payne & Ting, 2020). Similarly, variations in theta responses to perturbations with whole-body lean angle and perturbation magnitude (Stokkermans et al., 2022) have been taken to reflect continuous posture monitoring to facilitate an appropriate response to the threat imposed by a perturbation. Other studies have reported perturbation-evoked corticomuscular connectivity, suggesting that cortical activity may be functionally linked to the generation of balance-correcting muscle responses (Peterson & Ferris, 2019; Stokkermans et al., 2023). Finally, Boebinger et al. (2024) developed a neuro-mechanical model that predicted muscle activation from EEG signals, providing a framework in which EMG could

potentially be used to infer cortical contributions to balance control.

These interpretations, however, rest on a central, yet largely untested assumption that perturbation-evoked cortical responses reflect processes that are specific to, and contingent on, active balance control. This implies that the presence and characteristics of these cortical signals should depend on the postural relevance of the imposed perturbation. In this view, cortical activity is interpreted as contributing causally to the initiation or scaling of corrective motor actions. An alternative possibility is that these cortical responses do not invariably drive corrective output but instead are gated depending on postural relevance. Evidence from the vestibular system provides a useful parallel: vestibular contributions to muscle activity rapidly disengage when postural control is no longer required. For example, when vestibular error signals are artificially induced through electrical stimulation, post-ural muscle responses diminish if participants are fully supported and no balance correction is required (Britton et al., 1993; Fitzpatrick & McCloskey, 1994; Luu et al., 2012). If the N1 and theta signals reflect processes strictly coupled to balance control, then they too should be absent when postural control is disengaged. Testing this pre-mise is crucial to clarify whether these cortical signals are merely correlated with or are functionally linked to balance recovery.

This study aimed to directly assess whether balance N1 and theta responses depend on the nervous system's active engagement in postural control. To do this, we used a robotic balance simulator capable of mimicking and manipulating the physiological sensations and control of upright stance. In our first experiment, we compared the cortical responses to unexpected support-surface rotations under two conditions: (1) with participants actively balancing their body (Balance Control) and (2) with participants passively experiencing the same whole-body motion with comparable sensory signals and pre-perturbation motor engagement but without causal control over their movement (No-Control). Based on prior interpretations that causally link cortical markers to balance corrections, we hypothesized that these responses would diminish or disappear during passive conditions, in parallel with reduced muscle activity. Contrary to this hypothesis, cortical responses persisted in the No-Control condition despite substantial reductions in corrective muscle responses. While these findings indicate that the balance N1 and midfrontal theta activity are not tightly coupled to balance control or motor output, it is possible that the preservation of the N1 in the No-Control condition was supported by the close matching of sensory feedback and motor engagement preceding the perturbation. Therefore, we conducted a second experiment in which participants were passively moved under conditions that removed

whole-body sensory feedback (i.e. motion) and motor engagement (i.e. muscle activity) before the perturbation, thereby creating situations increasingly distant from balance control. We found that cortical responses were influenced only by sensory feedback, showing larger amplitudes when whole-body motion was absent, while lower-leg muscle responses were influenced only by motor engagement, decreasing when participants did not activate their muscles before the perturbation. Together, these results suggest that perturbation-evoked cortical activity is driven primarily by sensory input from the perturbation, rather than motor engagement or task relevance. Thus, cortical responses accompanying balance perturbations may reflect the brain's early detection and evaluation of unexpected sensory events.

## Materials and methods

### Participants

Twenty-two healthy adults were recruited from the Erasmus University Medical Centre and surrounding regions to participate in two separate experiments. Ten participants (five female, $25.3 \pm 6.9$ years, $1.76 \pm 0.12$ m, $68.2 \pm 11.0$ kg) completed Experiment 1 and 16 participants took part in Experiment 2 (eight female, $26.3 \pm 6.0$ years, $1.74 \pm 0.11$ m, $67.7 \pm 13.0$ kg), of whom four took part in both experiments. All participants had no self-reported history of neurological disorders or significant musculoskeletal injuries that may affect balance. The experimental protocol conformed to the *Declaration of Helsinki* and was approved by the Medical Research Ethics Committee Erasmus Medical Centre (NL76700.078.21, 14-04-2021). The protocol was explained before the experiment, and all participants provided written informed consent before participating.

### Experimental set-up

For all experiments, participants stood in a custom-designed robotic balance simulator that mimics the mechanics of natural standing balance when modelled as an inverted pendulum to control whole-body motion in anterior-posterior (AP) and mediolateral (ML) directions (Forbes et al., 2016; Huryn et al., 2010; Luu et al., 2012; Qiao et al., 2023; Rasman et al., 2024). The robotic system consisted of a backboard frame, hip and shoulder harnesses, and an ankle-tilt platform, each controlled by separate servo-motor-driven linear actuators (Fig. 1A). Additional details of the robot's mechanical design, including motors, actuators and mechanical linkages, can be found in Rasman et al. (2024), which has a similar design to the robot in Qiao et al. (2023). Briefly, the backboard frame controlled whole-body motion in

the AP direction, the hip and shoulder harnesses ensured movement in the ML direction, and the ankle-tilt platform controlled foot rotation. Participants stood barefoot on a force plate (AMTI BP400 × 600; Watertown, MA, USA) secured to the ankle-tilt platform, with their feet pointing anteriorly at hip joint width and ankles aligned with the robot's rotation axis (Sado et al., 2021). Participants were secured to the backboard with seatbelts attached to hip and shoulder harnesses at the level of their greater trochanter and sternal notch, respectively. The harnesses were lined with a medium-density foam and another layer of foam was placed between the participant and the seatbelts. The seatbelts prevented participants from falling anteriorly out of the device without supporting the vertical load of the body acting through the feet.

The dynamic model of the body was simulated using a real-time motion controller (PXI-8880; National Instruments, Austin, TX, USA) running at 500 Hz. Target encoder counts dictating motor positions in real-time were sent to an FPGA module (PXI-7846R; National Instruments) that directly communicated with the motors' servo drives. A detailed description of the inverted pendulum model used to simulate balance, as well as the control architecture of the system, can be found in Qiao et al. (2023) and Rasman et al. (2024). Briefly, the simulation was programmed as a concentrated mass (CoM) model to match each participant's physical dimensions, including body mass and height, CoM height, sternum height, trochanter width and ankle height. The CoM position was estimated by having participants lie supine on a rigid board that was balanced over a rod that was positioned transversely under the board. The board was rolled over the rod until the participant's mass was evenly distributed. The height of their CoM was measured from the ankle joint to the board's tipping point. When standing in the robotic balance simulator, participants were integrated into the robot's control loop, generating ankle torques to control the motion of their upright body through the robotic system (Fig. 1A). In the AP direction, the backboard and ankle-tilt platform rotated about an axis that passed through the ankle joints. In an anterior-leaning position, plantar–flexor torques applied to the force-plate were required to stabilize the system in the same way ankle torque muscles stabilize the body during normal standing. An increase in plantar–flexor torque greater than the torque created by gravity rotated the body posteriorly. In the ML direction, the simulator used a similar inverted pendulum model with its centre of rotation at the midpoint between the ankles. The simulation then distributed the angular position of the CoM to translation of the body at the level of the pelvis and shoulders for varying stance widths following mechanical models of upright standing that assume the pelvis remains perpendicular to the torso (Day et al., 1993; Goodworth & Peterka, 2012; Qiao et al., 2023). The hip and shoulder

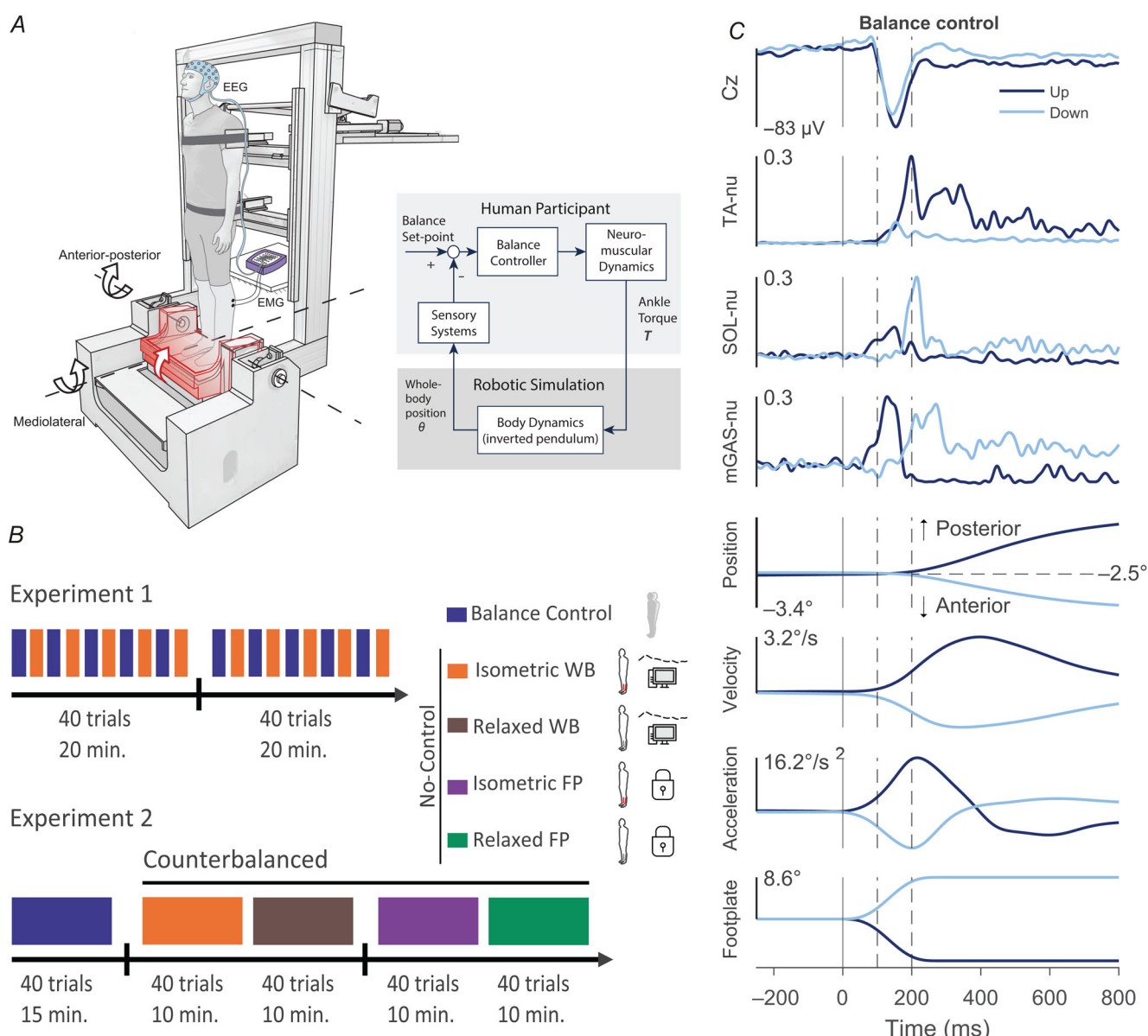

**Figure 1. Experimental set-up, timeline of the experimental protocol of Experiments 1 and 2, and sample data from Experiment 1**

*A*, participants stood on a force plate and were secured to a backboard using seatbelts attached to shoulder and hip harnesses. The force plate was secured to an ankle-tilt platform, which delivered support-surface perturbations that rotated along an axis aligned with the ankle joint. Perturbations were delivered in two directions (toes-down and toes-up); the schematic illustrates a toes-up perturbation. The balance control loop of the robotic balance simulator allowed participants to control their own balancing motion while exposed to support surface rotations. Based on the torque exerted by the participants onto the force plate, the backboard rotated the whole-body in the anterior–posterior direction and the hip and shoulder harnesses moved the whole-body in the mediolateral direction. *B*, experimental protocol of Experiment 1 (top) and Experiment 2 (bottom). Both experiments included a Balance Control condition. Experiment 1 consisted of 80 trials (40 per direction), arranged in alternating blocks of five Balance Control and five No-Control trials. The No-Control condition from Experiment 1 corresponds to the isometric whole-body condition in Experiment 2. In Experiment 2, each No-Control condition consisted of 40 trials (20 per direction) and conditions were counterbalanced across participants. No-Control conditions are illustrated with two icons: the first indicates the plantarflexor muscle activity prior to the perturbation (isometric contraction or relaxed), and the second indicates whole-body sensory feedback, which included either prerecorded whole-body movement controlled by a computer or no whole-body motion (footplate + whole-body (WB) or footplate-only (FP)). Breaks in between conditions (black vertical lines) took 5–10 min. *C*, data from a representative subject from Experiment 1 averaged over all trials during Balance Control conditions. From top to bottom: Cz electrode EEG and

EMG activity, whole-body position, velocity and acceleration, and angle of the footplate. Positive going whole-body position indicates backward movement; positive going footplate angle indicates toes-down perturbations. The darker line represents the toes-up direction, and the lighter line represents the toes-down direction. The solid vertical lines represent perturbation onset. The spaces between dashed vertical lines represent the window during which the N1 response was quantified (100–200 ms).

harnesses were supported by separate gas springs minimizing vertical load on the participant.

Angular motion of the body was constrained using software limits to ensure participants could generate enough torque to correct their balance across the range of motion (AP: 6° anterior to 3° posterior, ML: 3° left to 3° right). When participants exceeded these limits, a supportive torque was gradually increased (linear over a range of 1° beyond the simulated balance limits) such that they eventually could not rotate further in that direction regardless of the torques they applied. A damping term over this same range ensured a smooth attenuation of motion. Participants could autonomously get out of these limits and continue balancing by generating active torques in the opposite direction.

## Experimental protocol

**Familiarization and baseline.** Participants in both experiments first underwent a familiarization period and a baseline balance trial to estimate several parameters used to conduct the experiment. When first secured to the robot, participants were aligned vertically to minimize offset forces and torques on the force plate. First, the participant's ankle centres of rotation (anterior edge of the medial malleolus; Sado et al., 2021) were aligned with the backboard axis of rotation and at an equal distance from the midline. We then adjusted the depth of the hip and shoulder harnesses to ensure the ankle AP and ML moments were within $\pm10$ Nm and the horizontal forces were within $\pm10$ N while ensuring participants were in a comfortable vertical posture. Pieces of tape placed on the force plate ensured consistent foot placement throughout the experiment.

Before any experiment started, all participants underwent a familiarization session to become acquainted with the balance simulator and the balancing task. During this session, participants were instructed to sway back and forth and left to right, allowing the balance simulator to reach its position limits (6° anterior, 3° posterior, 3° left, 3° right). When the limits were exceeded, participants were instructed to generate ankle torque in the opposite direction to restore their balance. Once participants were accustomed to standing on the robot and could maintain upright posture with ease, they practised responding to the perturbations until they could maintain balance without falling into the limits for at least two consecutive trials in both directions (see 'General procedures'). The familiarization session lasted approximately 5 min.

Participants then performed a baseline balance trial where they were instructed to stand at their preferred angle (Experiment 1: $-0.81 \pm 0.56°$; Experiment 2: $-1.48 \pm 0.76°$) for 1 min. From this recording, we calculated the variability (i.e. SD) of their applied ankle torque (Experiment 1: $3.40 \pm 1.4$ Nm; Experiment 2: $3.62 \pm 1.4$ Nm) and whole-body velocity (Experiment 1: $0.18 \pm 0.04°$/s; Experiment 2: $0.18 \pm 0.05°$/s) to determine when perturbations should be delivered (see 'General procedures').

**General procedures.** The robot delivered ankle perturbations by rotating the ankle-tilt platform (or footplate) toes-up or toes-down with a ramp displacement of 0.15 radians within 0.3 s (peak velocity: 62.7°/s; peak acceleration 717.3°/s$^2$). A toes-down rotation resulted in anterior whole-body motion and required ankle plantarflexion torque to return to an upright position, while a toes-up rotation resulted in posterior whole-body motion and required an ankle dorsi-flexion torque to ensure balance. These perturbations were designed according to the minimum-jerk profile described by Vlutters et al. (2015), with zero acceleration at both the start and end of the perturbation to ensure smooth rotations.

To minimize recording artefacts in the EEG data throughout the perturbation period, participants were asked to refrain from blinking or contracting facial/throat muscles once they were at the correct whole-body orientation (see target description below) until ~2–3 s after receiving the perturbation. Perturbations were manually initiated by the experimenter during periods where EEG activity showed no obvious artefacts (i.e. driven by eye movements, blinks, etc.), as determined through visual inspection of real-time data. All perturbations were delivered unpredictably by randomizing their timing and direction, with a minimum time of 16 s between each perturbation. Participants typically required about 5–15 s to recover their balance following the perturbation and return their whole-body angle to the target, resulting in an average interval of $26 \pm 5$ s between perturbation onsets.

**Experiment 1.** In Experiment 1 ($n = 10$), we tested two conditions to assess cortical responses to support-surface rotations: the Balance Control condition, where participants actively controlled their balance, and the No-Control condition, where participants had no control

over their balance movement and passively experienced the footplate and whole-body motion recorded during the balance trials (Fig. 1*C*, Experiment 1). In the Balance Control condition, participants were instructed to hold a steady whole-body position at the anterior target angle of 2.5° while they maintained upright balance on the robot. Support-surface perturbations were delivered through the footplate when the participant's whole-body angle was within $1.5 \times$ SD of the target angle (2.5° anterior) and their whole-body velocity was below $2 \times$ SD; these ranges were calculated from the balance behaviour during their baseline trial. Participants received real-time visual feedback of their whole-body position on a screen placed ∼1.5 m in front of them. Whole-body centre of mass position was presented as a red circle on a black background that turned green when their position and velocity were within the predefined target ranges. Provided the target was green, the experimenter delivered the perturbation at a random interval between 1 and 8 s. At the onset of the footplate rotation, the visual feedback disappeared, leaving a black screen.

In the No-Control condition, participants were initially oriented at a fixed whole-body angle of 2.5° anterior prior to the onset of the support-surface perturbation. They were informed that they would have no control over their motion and that the support-surface and their whole body would move according to prescribed motion profiles (Leeuwis et al., 2024). These movement profiles were identical to those recorded during the Balance Control condition – but randomized in their order – ensuring that any differences observed in cortical and muscle responses were not due to variations in experienced motion. Before a perturbation was delivered, participants generated a plantarflexion torque at a level equivalent to the torque needed to maintain a whole-body angle of 2.5° anterior during the Balance Control trials (i.e. $T = 0.0436$ mg/l). Perturbations were delivered when the participant's torque was within $1.5 \times$ SD of the target torque, where the SD was derived from the baseline trials. Visual feedback of the ankle torque was presented to the participant in a comparable manner to that used in the Balance Control condition (i.e. a red circle that turned green when torque was within the predefined limits). Before the first No-Control trial, participants were given a single perturbation to familiarize themselves with the experience of the movement and to understand that they did not have any control over their movement during these trials. In the No-Control condition, muscle activity was visually monitored, and when sustained muscle contractions were observed during the trials, participants were reminded that these had no influence on their motion throughout the trials.

Experiment 1 consisted of 80 trials in total, with Balance Control and No-Control conditions alternating after every five trials. This repeated block design was chosen to mitigate habituation effects, which are known to diminish balance N1 responses over repeated trials (Mierau et al., 2015; Payne, Hajcak, et al., 2019). However, within each group of 10 trials, the Balance Control conditions were always performed first in order to generate the motion profiles driving participant movement during the No-Control conditions. All trials were evenly distributed between both perturbation directions (i.e. 40 toes-up rotations and 40 toes-down rotations). After 40 trials, participants were given a 5–10 min break off the robot (see Fig. 1*B*, Experiment 1), during which they could sit or walk around the room.

**Experiment 2.** In Experiment 2 ($n = 16$), we investigated whether manipulating the sensory feedback and/or motor engagement modulated the cortical N1 responses during our No-Control condition (Fig. 1*B*, Experiment 2). By changing the sensory and motor signals accompanying the perturbations, we aimed to test whether cortical responses differ as the conditions progressively deviate from the active balance trials. Participants were first exposed to support-surface rotations during the Balance Control condition, reproducing the set-up from Experiment 1. Participants then underwent four different No-Control conditions, each designed to vary sensory feedback and/or motor engagement. All No-Control conditions were performed separately, with participants positioned at a fixed whole-body angle of 2.5° anterior at the beginning of each trial. Sensory feedback was modulated by exposing participants to either footplate and whole-body rotations (i.e. whole-body) or footplate-only rotations. For whole-body trials, the footplate and whole-body motion profiles replicated those recorded during the Balance Control condition [see Leeuwis et al. (2024) for technical implementation]. In footplate-only trials, only the footplate moved, imposing support-surface rotations without whole-body motion. Motor engagement was manipulated by instructing participants to perform an isometric contraction against the footplate or remain relaxed prior to the perturbation. In the isometric conditions, they generated plantarflexion torques equal to those required to maintain upright standing at the anterior whole-body angle of 2.5°.

We opted for participants to complete all trials in the Balance Control condition first, rather than using the repeated block design from Experiment 1, due to time necessary to process the recorded motion data and complete all four No-Control conditions. To mitigate the known effects of habituation that can occur from repeated perturbation exposure (Mierau et al., 2015; Payne, Hajcak, et al., 2019), we analysed responses exclusively across the No-Control trials. Additionally, to further reduce the impact of habituation, the order of the No-Control conditions was different for each

participant. The experiment consisted of 200 trials in total, with 40 trials per condition evenly distributed between both directions (i.e. 20 toes-up rotations and 20 toes-down rotations). After the Balance Control trials, participants took a 5–10 min break off the robot. During the subsequent No-Control conditions, participants were given another break off the robot after completing two conditions.

### Data acquisition and analysis

Data were processed using custom-designed scripts in MATLAB software (2022b version, Mathworks, Natick, MA, USA) and the EEGLAB toolbox (v2022.0, Delorme & Makeig, 2004). Group data in the text and figures are presented as mean ± SEM unless otherwise specified. Statistical analyses were performed using JASP (Version 0.19.1), with a predefined significance level of 0.05.

**Electroencephalography.** EEG signals were obtained using a 64-channel electrode cap (Infinity headcap, TMSi, Oldenzaal, The Netherlands) based on the International 10–20 System. Active electrode sites were prepared by applying a conductive electrode gel (Electro-Gel, Electro-Cap International, Eaton, OH, USA) using a blunt-tip needle, which was simultaneously used to rub the scalp to improve electrode impedance. Impedances of Cz and surrounding frontal, central electrodes (CPz, CP1, CP2, C1, C2, FCz, FC1, FC2, Fz, F1, F2) were controlled to be below 10 kΩ before the start of data collection. For all other electrodes, the impedances were controlled to be below 20 kΩ. EEG signals were visually checked for noisy or flat line channels before data collection and electrode corrections were made accordingly. EEG data were amplified on a 128-channel stationary amplifier system (REFA amplifier, TMSi) with actively shielded electrode cables to limit artefacts. Data were sampled at a frequency of 2048 Hz and stored on a laptop via a fiberoptic-to-USB converter (Fusbi) for offline analysis. To further limit motion artefacts, the EEG cable bundle was secured to the robotic simulator at the shoulder and hip supports, and the amplifier was placed close behind the participant at the approximate height of the knees.

Raw EEG data were high-pass filtered offline at 1 Hz to attenuate slow drifts and motion-related contamination, and low-pass filtered at 40 Hz using a zero-phase, finite impulse response (FIR) filter implemented using EEGLAB's *eegfiltnew* function (v2022, Delorme & Makeig 2004). Channels with flatline periods lasting over 5 s and those with a correlation with nearby channels of 0.77 or lower were identified and removed. Subsequently, the data were sectioned into 2.5 s segments starting 1 s before perturbation onset (i.e. time = 0). Single-trial epochs were then baseline-corrected by subtracting the mean voltage from each channel calculated from 150 to 50 ms before perturbation. Finally, independent component analysis (ICA) was performed to remove artefactual components from the data (i.e. eye blinks and muscle activity; pop runica algorithm; Delorme & Makeig, 2004). Additionally, components were inspected visually and were removed when they reflected clear non-neural artefacts, including motion-related activity, line noise, cardiac signals or channel noise (occurring in only one participant in one condition where the first component was clearly not arising from brain activity). Single-subject cortical ERPs were computed separately for each condition by averaging EEG electrode data across all trials within that condition. As the balance N1 is typically largest at the Cz electrode (see Fig. 2), we used this signal to quantify the peak ERP magnitude between 100 and 200 ms after perturbation onset. The latency of the N1 response was taken as the time of the peak relative to perturbation onset.

To examine how midfrontal theta activity varied across the different conditions, we applied a data-driven spatial filtering approach designed to extract the components that maximize theta power. Specifically, we used generalized eigendecomposition (GED) on the cleaned EEG data (i.e. after ICA) using established methods (Cohen, 2017). GED was performed using the covariance matrices of theta-filtered (3–8 Hz) and broadband EEG data pooled across all conditions. This yielded a set of eigenvectors (channel weights) with corresponding eigenvalues. The eigenvector with the largest eigenvalue was identified as the spatial filter that maximizes energy in the theta band. For each participant, we selected the eigenvector with the largest eigenvalue that also displayed a clear ERP average and a midfrontal/midline scalp topography, as the spatial filter for performing further analysis (Cohen, 2022).

This spatial filter was then used to compute the midfrontal theta component's time series for each condition by performing a dot product between the spatial filter (a vector of channel weights) and each condition's EEG data matrix (channels by time), resulting in a weighted sum of channel activity. To visualize spectral dynamics over time, we applied a time–frequency decomposition to these filtered single-trial data using a complex Morlet wavelet convolution. We used 40 frequencies linearly spaced between 2 and 60 Hz, with wavelet widths logarithmically spaced from four to 12 cycles (Stokkermans et al., 2022). Theta power in the midfrontal components was quantified by bandpass filtering the individual timeseries using consecutive 3 Hz high-pass and 8 Hz low-pass third-order Butterworth IIR filters, followed by a Hilbert transform to compute the signal envelope. The Hilbert transform was applied over the whole trial time series of 2.5 s, ensuring that the duration of the signal was sufficiently long to compare theta power over a whole wave cycle. Data were then averaged across trials, and the maximum magnitude of this trial average signal in the 100–300 ms window

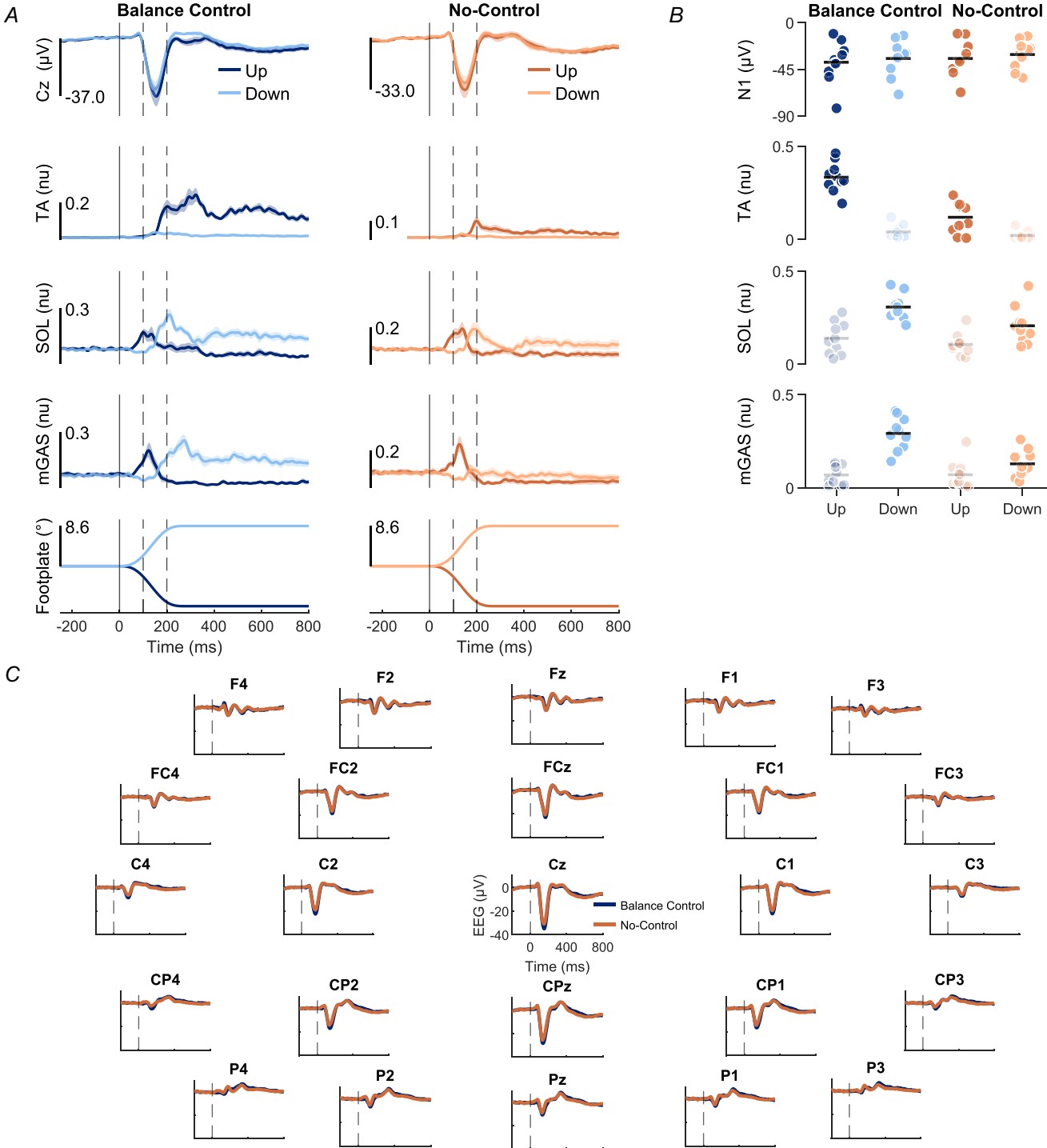

**Figure 2. Group mean data from Experiment 1 of EEG activity and TA, SOL and mGAS EMG activity**

*A*, average Cz ERPs and EMGs over participants (*n* = 10) for Balance Control and No-Control conditions. The darker line represents the toes-up direction and the lighter line represents the toes-down direction. The solid vertical lines represent perturbation onset. The spaces between dashed vertical lines represent the window in which the N1 response was observed (i.e. 100-200 ms). For each muscle, only trials where that muscle was relevant for balance correction were included for analysis: toes-up perturbations for TA responses and toes-down perturbations for SOL/mGAS responses. Furthermore, EMG analysis focused on the corrective phase of activity, from 200 to 400 ms after perturbation onset (after the second dashed line). EMG signals for each muscle were normalized to the maximum recorded value across all trials. *B*, Cz EEG and EMG amplitudes in both perturbation directions for Balance Control and No-Control conditions. Each dot represents a participant, and horizontal bars represent the group mean of the amplitude. Faded markers represent data from the non-relevant perturbation direction for

each muscle. *C*, average ERPs of all recorded scalp-electrodes over participants for Balance Control and No-Control conditions in the toes-up direction.

after perturbation onset was extracted as peak theta power for each condition per participant. This approach ensures that the estimated theta power accurately reflects the ongoing oscillatory activity in that window, even if theta oscillations are associated with longer-lasting events (Stokkermans et al., 2022).

**Electromyography.** Surface EMG was collected from three muscles of the right lower limb: the tibialis anterior (TA), medial gastrocnemius (mGAS) and soleus (SOL). These muscles were selected because of their roles as primary agonist (mGAS and SOL) and antagonist (TA) muscles in balancing in the AP direction (Di Giulio et al., 2009). An additional ground electrode was placed on the right wrist. We used self-adhesive Ag–AgCl surface electrodes (Blue Sensor M, Ambu A/S, Ballerup, Denmark) with an inter-electrode distance of 2 cm. Before placing the electrodes, the skin was cleaned with an iso-propyl alcohol wipe and scrubbed with a skin preparation gel (Nu-Prep, Weaver and Company, Aurora, CO, USA). After placing the electrodes, an additional piece of tape was added to better secure them to the skin. EMG signal quality was visually checked by asking participants to dorsiflex and plantarflex their ankle prior to any data collection. The EMG signals were sampled concurrently with the EEG data at 2048 Hz using the same amplifier.

Raw EMG data were offline high-pass filtered at 10 Hz (zero-phase FIR filter, no lag, order determined by a 2 Hz transition band width), rectified, and then low-pass filtered at 30 Hz (zero-phase FIR filter, no lag, order determined by a 7.5 Hz transition band width). Single-trial EMG data were normalized to 1 using the maximum value recorded from all trials for each participant. EMG time traces were then averaged across trials within each condition and direction for each participant. Muscle response amplitudes were extracted over two latency windows following perturbation onset: a short latency period (0–200 ms) reflecting initial antagonist activity evoked by the perturbation, and a longer-latency period (200–400 ms) capturing balance-correcting muscle responses.

**Kinematics.** Throughout the experiments, we estimated the angular position of the body centre of mass using encoders in the hip, shoulder and backboard motors. From this whole-body centre of mass angle, we also estimated the whole-body angular velocity and acceleration by taking the first and second derivative of the angle. These kinematic measures from baseline trials were used to define the target range that participants were required to maintain prior to perturbation (see

'Experimental protocol'). Further, kinematic signals measured throughout the perturbation trials in the Balance Control condition were used to prescribe the movement profiles used during the No-Control conditions.

## Statistical analysis

**Experiment 1.** To assess whether corrective muscle activity differed across conditions, we performed pairwise *t*-tests on both the initial peak EMG responses (0–200 ms) and the balance-correcting peak EMG responses (200–400 ms) of each muscle. For the initial muscle responses, we included only the subset of trials in which the perturbation stretched the recorded muscle, capturing its short-latency stretch response. Specifically, TA responses were analysed during toes-down perturbations and gastrocnemius/soleus responses only during toes-up perturbations. These responses were expected to be driven by initial sensory input and would not differ between conditions. For the balance-correcting muscle activity, we included only the subset of trials in which the perturbation direction required that muscle to contribute to balance corrections (i.e. TA during toes-up perturbations, and mGAS/SOL during toes-down perturbations). We expected the balance-correcting muscles responses would be reduced in the No-Control condition compared to the Balance Control condition. To assess whether the balance N1 depends on active balance control, we performed repeated-measures ANOVAs on the balance N1 amplitudes and latencies recorded from the Cz electrode in our Balance Control and No-Control conditions. This analysis included two independent variables: condition (Balance Control *vs*. No-Control) and perturbation direction (toes-up *vs*. toes-down). Additionally, we conducted a similar repeated-measures ANOVA on peak theta power.

**Experiment 2.** To test whether sensory and motor conditions affected the muscle activity that is normally expected to correct for balance (200–400 ms after perturbation onset), we applied 2 × 2 repeated-measures ANOVAs to peak EMG responses in each muscle. The within-subject factors were sensory feedback (footplate + whole-body *vs*. footplate-only) and motor engagement (isometric contraction *vs*. relaxed). The analyses only considered the subset of trials where the perturbation would have required that muscle's contribution. Here, we did not assess the initial muscle responses (0–200 ms) as we found no difference across Balance Control and No-Control conditions in

Experiment 1 (see Results). To determine whether the sensory or motor signals present during the imposed motion influenced cortical activity in the No-Control conditions, we conducted $2 \times 2 \times 2$ repeated-measures ANOVAs on the amplitude and latency of the balance N1. The independent variables were sensory feedback (footplate + whole-body *vs.* footplate-only), motor engagement (isometric contraction *vs.* relaxed) and perturbation direction (toes-up *vs.* toes-down). A $2 \times 2$ repeated-measures ANOVA with sensory feedback and motor engagement as within-subject factors was performed on the peak midfrontal theta power to assess effects on oscillatory activity, with data combined across toes-up and toes-down perturbations given the absence of directional effects in the N1 analysis (see 'Results'). For all results, descriptive statistics are reported as mean ± SEM to indicate precision of the group estimate. For percentage differences, values are reported as mean ± SD to reflect inter-individual variability.

## Results

### Active balance control alters corrective muscle activity with limited changes in cortical responses

In Experiment 1 ($n = 10$), participants experienced unexpected toes-up and toes-down support-surface rotations in the robotic balance simulator under two conditions (Balance Control and No-Control; Fig. 1*B*, Experiment 1). Participants stood at a 2.5° anterior whole-body angle and completed 40 trials per condition, delivered as eight blocks of five perturbations per condition, with Balance Control always preceding No-Control. In the Balance Control condition, toes-up perturbations accelerated the whole-body CoM posteriorly, requiring a corrective dorsiflexion torque to decelerate the body and maintain upright posture (see Fig. 1*C* for a representative participant). In contrast, toes-down perturbations accelerated the CoM anteriorly, necessitating a corrective plantarflexion torque to stabilize the body. Whole-body displacements peaked at $2.0 \pm 0.6°$ posterior for toes-up and $1.5 \pm 0.4°$ anterior for toes-down perturbations (see Fig. 1*C*). In the No-Control condition, participants were fixed and maintained a plantarflexion torque equal to the static load of standing before being passively exposed to the perturbation and whole-body motion experienced during the Balance Control trials.

In the Balance Control condition, toes-up perturbations evoked initial EMG responses in mGAS and SOL muscles (0–200 ms), likely driven by early sensory feedback from the perturbation (Carpenter et al., 1999; Henry et al., 1998; Nashner, 1976; see Fig. 2*A*). These responses were followed by a balance-correcting burst of TA activity, which peaked over ∼200–400 ms and provided the dorsiflexion torque needed to counteract

the perturbation. Finally, we observed longer-latency stabilizing activity (∼400–800 ms), which served to maintain equilibrium at the new whole-body orientation (Carpenter et al., 1999). Toes-down perturbations elicited the opposite pattern over a similar time course, with an initial sensory-driven peak in TA activity followed by balance-correcting and stabilizing activity in mGAS and SOL (see Fig. 2*A*). In the No-Control condition, the initial EMG responses were unchanged relative to the Balance Control condition (TA down: Balance Control $0.05 \pm 0.01$ *vs.* No-Control $0.04 \pm 0.01$, $P = 0.364$, $d = 0.30$ [95% CI: –0.340 to 0.93]; SOL up: $0.23 \pm 0.04$ *vs.* $0.24 \pm 0.04$, $P = 0.651$, $d = -0.15$ [–0.77 to 0.48]; MG up: $0.24 \pm 0.04$ *vs.* $0.28 \pm 0.05$, $P = 0.071$, $d = -0.65$ [–1.32 to 0.05]). In contrast, the balance-correcting peak muscle responses were markedly reduced (see Fig. 2*B*). Specifically, during toes-up perturbations, the peak balance-correcting TA activity decreased by $66.4 \pm 22.6\%$ (Balance Control: $0.33 \pm 0.03$ *vs.* No-Control: $0.12 \pm 0.03$; $t = 9.31$, $P = 6.472 \times 10^{-6}$, $d = 2.94$ [95% CI: 1.46–4.40]). During toes-down perturbations, peak SOL activity decreased by $33.8 \pm 22.0\%$ (Balance Control: $0.31 \pm 0.02$ *vs.* No-Control: $0.21 \pm 0.03$; $t = 4.60$, $P = 0.001$, $d = 1.45$ [0.53–2.34]) and mGAS activity decreased by $53.7 \pm 21.9\%$ (Balance Control: $0.29 \pm 0.03$ *vs.* No-Control: $0.13 \pm 0.02$; $t = 4.79$, $P = 9.926 \times 10^{-4}$, $d = 1.51$ [0.57–2.42]). These findings demonstrate that longer-latency, balance-correcting muscle activity is substantially reduced when participants are no longer required to stabilize their body, confirming that corrective EMG responses are tightly linked to active postural control.

To assess whether N1 responses to perturbations differed across conditions, we analysed EEG activity at the Cz electrode, testing the hypothesis that cortical responses depend on active balance control and diminish when participants are passively moved. In the Balance Control condition, a robust N1 response was observed in all participants between 100 and 200 ms for both perturbation directions, confirming that our perturbation paradigm reliably elicited cortical activity during active balancing in the robot (see Fig. 2*A*). The N1 peaked at an average amplitude of $-38.2 \pm 6.4$ and $-34.5 \pm 5.5$ μV during toes-up and toes-down perturbations, with latencies of $148.0 \pm 3.2$ and $153.3 \pm 2.9$ ms, respectively (Table 1). In the No-Control condition, the N1 response persisted despite the absence of balance control, with only modest reductions in amplitude. On average, N1 amplitudes decreased by $9.02 \pm 6.83\%$ (toes-up: $-34.5 \pm 5.4$ μV; toes-down: $-30.7 \pm 4.3$ μV) compared to the Balance Control condition, resulting in a significant main effect of condition [$F_{(1,9)} = 7.08$, $P = 0.026$, $\eta^2 = 0.17$] but no significant effect of direction [$F_{(1,9)} = 3.60$, $P = 0.090$, $\eta^2 = 0.17$], and no interaction between condition and direction [$F_{(1,9)} = 0.06$,

**Table 1. Cz N1 and longer-latency (200–400 ms) muscle responses. EMG values for non-relevant perturbation directions are grey**

| Exp | Condition | | | Direction | N1 latency (ms) Mean | SD | N1 amplitude (µV) Mean | SD | TA amplitude (nu) Mean | SD | SOL amplitude (nu) Mean | SD | mGAS amplitude (nu) Mean | SD |
|---|---|---|---|---|---|---|---|---|---|---|---|---|---|---|
| 1 | Balance Control | | | Up | 148.0 | 10.05 | −38.15 | 20.14 | 0.33 | 0.08 | 0.14 | 0.09 | 0.07 | 0.05 |
| | | | | Down | 153.3 | 9.03 | −34.54 | 17.42 | 0.04 | 0.04 | 0.31 | 0.07 | 0.29 | 0.09 |
| | No-Control | | | Up | 149.5 | 15.78 | −34.53 | 17.16 | 0.12 | 0.08 | 0.11 | 0.06 | 0.07 | 0.07 |
| | | | | Down | 149.0 | 11.44 | −30.69 | 13.59 | 0.02 | 0.02 | 0.21 | 0.10 | 0.13 | 0.07 |
| 2 | Balance Control | | | Up | 150.3 | 7.90 | −30.18 | 15.59 | 0.34 | 0.10 | 0.11 | 0.05 | 0.07 | 0.03 |
| | | | | Down | 146.6 | 9.76 | −31.70 | 12.16 | 0.05 | 0.05 | 0.27 | 0.08 | 0.28 | 0.11 |
| | No-Control | Whole-body | Isometric | Up | 142.7 | 12.80 | −22.49 | 13.65 | 0.05 | 0.05 | 0.09 | 0.05 | 0.07 | 0.05 |
| | | | | Down | 139.3 | 13.79 | −24.79 | 11.00 | 0.02 | 0.02 | 0.14 | 0.07 | 0.09 | 0.06 |
| | | | Relaxed | Up | 142.4 | 9.18 | −24.87 | 12.59 | 0.05 | 0.07 | 0.07 | 0.04 | 0.03 | 0.03 |
| | | | | Down | 141.9 | 9.91 | −25.97 | 11.22 | 0.02 | 0.04 | 0.11 | 0.08 | 0.05 | 0.07 |
| | | Footplate only | Isometric | Up | 149.4 | 11.11 | −28.73 | 14.32 | 0.05 | 0.06 | 0.10 | 0.06 | 0.07 | 0.07 |
| | | | | Down | 142.7 | 13.85 | −27.31 | 13.63 | 0.03 | 0.06 | 0.14 | 0.08 | 0.08 | 0.05 |
| | | | Relaxed | Up | 149.5 | 9.82 | −28.20 | 12.06 | 0.03 | 0.03 | 0.07 | 0.06 | 0.04 | 0.05 |
| | | | | Down | 146.9 | 11.92 | −29.26 | 12.32 | 0.02 | 0.02 | 0.12 | 0.09 | 0.03 | 0.05 |

$P = 0.816$, $\eta^2 = 1.47 \times 10^{-4}$]. N1 latencies were similar in the No-Control condition (toes-up: 149.5 ± 5.0 ms; toes-down: 149.0 ± 3.6 ms), with no effect of balance conditions [$F_{(1,9)} = 0.43$, $P = 0.528$, $\eta^2 = 0.01$] or perturbation directions [$F_{(1,9)} = 0.72$, $P = 0.418$, $\eta^2 = 0.04$], and no interaction between condition and direction [$F_{(1,9)} = 4.81$, $P = 0.056$, $\eta^2 = 0.06$; see Table 1]. Finally, a qualitative analysis of 24 surrounding electrodes showed a consistent pattern with the responses at Cz: N1 responses remained clearly visible with similar timing in the No-Control conditions, supporting the persistence of early cortical processing despite the absence of active postural control (see Fig. 2C).

To further assess any changes in cortical activity across conditions, we also examined midfrontal theta responses (4–8 Hz), combining toes-up and toes-down perturbations based on the absence of directional effects in the N1 analysis. We applied GED to each participant's EEG data to identify spatial filters optimized for theta band activity (see 'Methods'). The resulting components consistently exhibited midline scalp topographies across participants (Solis-Escalante et al., 2019), demonstrating that theta activity evoked by postural perturbations is best captured at midline electrodes (see Fig. 3A). Time–frequency analysis of the extracted theta component revealed a clear increase in theta power following perturbation onset in both conditions, peaking at latencies similar to the N1 (∼150 ms). This increase was slightly greater in the Balance Control condition compared to the No-Control condition (see Fig. 3B). To quantify this difference, we band-pass filtered the component time series in the theta range (4–8 Hz) and extracted the peak in theta power using the Hilbert transform. Across participants, peak theta power decreased by 11.3 ± 11.7% in the No-Control condition (2.3 ± 0.1 µV) relative to the Balance Control condition [2.1 ± 0.2 µV; $t_{(9)} = 2.71$, $P = 0.024$, $d = 0.86$ [95% CI: 0.11–1.57]] (see Fig. 3C). Combined with the N1 data, these results demonstrate that active balance control substantially influences longer-latency, balance-correcting muscle responses to perturbations, while early cortical responses – in both evoked potentials and oscillatory activity – are largely preserved across Balance Control and No-Control conditions. This indicates that these cortical responses occur irrespective of whether participants were actively controlling posture.

## Cortical responses to perturbations persist when sensory feedback and motor engagement are removed

Experiment 1 showed that the N1 persists without active balance control. To test whether this persistence depended on matched sensory feedback and motor engagement

preceding the perturbation, Experiment 2 independently manipulated sensory feedback (footplate + whole-body *vs.* footplate-only) and motor engagement (isometric *vs.* relaxed), yielding four No-Control conditions in addition to Balance Control (see 'Methods'). Balance Control trials were completed first to record the whole-body motion profiles replayed in the relevant No-Control conditions. Qualitatively, both the muscle activity and evoked cortical responses in the equivalent Balance Control and No-Control conditions of Experiments 1 and 2 were comparable (see Table 1 and Fig. 4).

Across the No-Control conditions, we first observed that sensory feedback (footplate + whole-body *vs.*

footplate-only perturbations) had no significant effect on the balance-correcting EMG peak response (i.e. from 200 to 400 ms) for any muscle [TA toes-up: $F(1,15) = 1.54$, $P = 0.234$, $\eta^2 = 0.02$; SOL toes-down: $F(1,15) = 0.10$, $P = 0.762$, $\eta^2 = 0.001$; mGAS toes-down: $F(1,15) = 0.89$, $P = 0.361$, $\eta^2 = 0.02$; see Table 1]. In contrast, motor engagement significantly increased most muscle responses, with larger peak responses when participants maintained an isometric contraction as compared to relaxed conditions. This effect was observed for SOL [$F(1,15) = 11.83$, $P = 0.004$, $\eta^2 = 0.25$] and mGAS [$F(1,15) = 14.48$, $P = 0.002$, $\eta^2 = 0.30$], with responses increasing on average by $54.2 \pm 71.0$ and $314.5 \pm 523.9\%$,

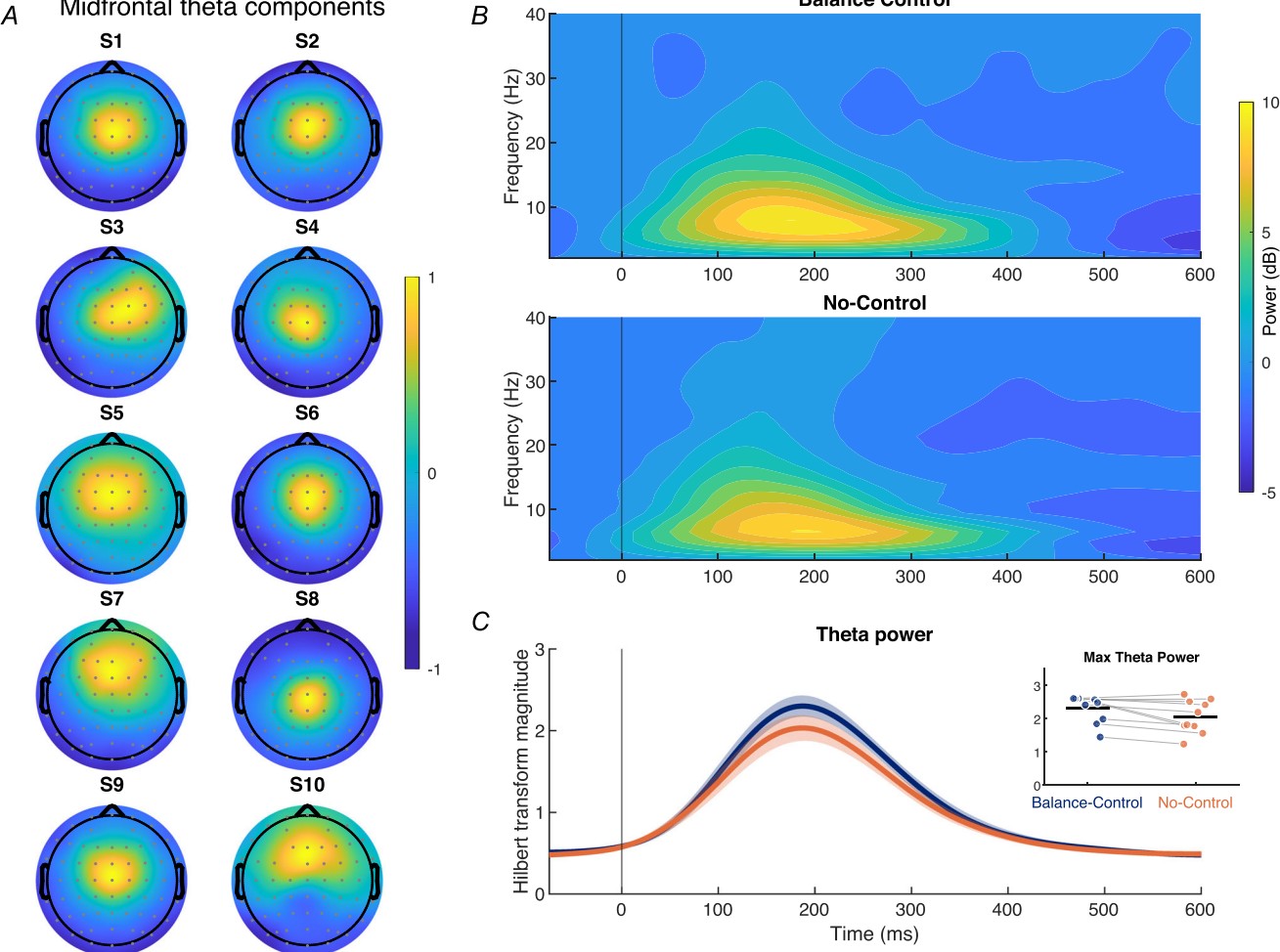

**Figure 3. Midfrontal theta modulation in Experiment 1**
*A*, participant-specific scalp topographies of the theta component identified from all trials (combining Balance Control and No-Control conditions) via generalized eigendecomposition (GED), showing a consistent mid-line distribution across participants. Topographies were normalized within participants. *B*, time–frequency representations of power from the identified theta component, averaged across participants. The upper panel shows responses during the Balance Control condition, and the lower panel shows responses during the No-Control condition. Both conditions exhibit an increase in theta-band power following perturbation onset, with a stronger increase in the Balance Control condition. *C*, time series of the Hilbert transform magnitude within the theta band, averaged across participants, for Balance Control (blue) and No-Control (orange) conditions. The inset shows individual participants' peak theta power in each condition.

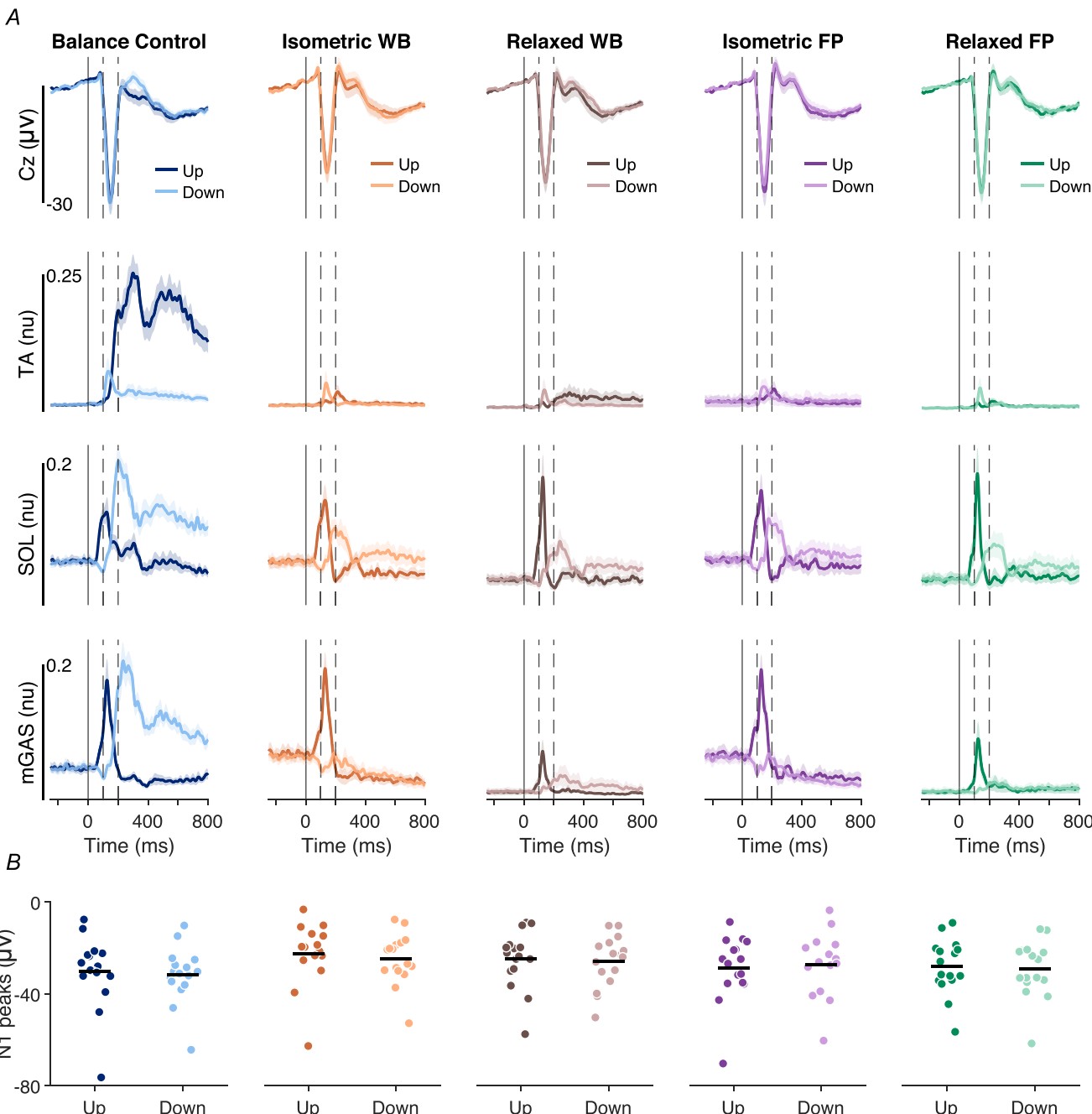

**Figure 4. Group mean data from Experiment 2 of Cz electrode EEG activity and TA, SOL and mGAS EMG activity**

*A*, average ERPs and EMGs over participants (*n* = 16) for Balance Control and No-Control conditions: isometric whole-body (same as No-Control in Experiment 1), relaxed whole-body, isometric footplate-only and relaxed footplate-only. The darker line represents the toes-up direction, and the lighter line represents the toes-down direction. The solid vertical lines represent perturbation onset. The spaces between dashed vertical lines represent the window during which the N1 response was observed. For each muscle, only trials where that muscle was relevant for balance correction were used for analysis: toes-up perturbations for TA responses and toes-down perturbations for SOL/mGAS responses. EMG analysis focused on the corrective phase of activity, from 200 to 400 ms after perturbation onset. *B*, N1 peak amplitudes in both directions for all conditions. Each dot represents a participant (*n* = 16) and horizontal bars represent the group mean of the amplitude. Abbreviations: WB, footplate + whole-body; FP, footplate-only.

respectively, but not for TA [$F(1,15) = 1.03$, $P = 0.327$, $\eta^2 = 0.03$]. Together, these results indicate that in the absence of balance control, muscle activity depends more on the preceding motor engagement than on the presence or absence of whole-body movement.

Initial assessment of EEG activity at the Cz electrode showed a clear N1 response in all No-Control conditions, but the changes observed across conditions did not parallel the modulation of muscle activity by motor engagement (Fig. 4). N1 amplitudes were significantly affected by sensory feedback [$F(1,15) = 14.99$, $P = 0.002$, $\eta^2 = 0.22$], with amplitudes in footplate-only conditions ($-28.4 \pm 3.3$ μV) being $13.4 \pm 12.8\%$ larger compared to conditions involving both footplate and whole-body motion ($-24.5 \pm 3.0$ μV, see Fig. 4). By contrast, there was no significant main effect of motor engagement [$F(1,15) = 2.72$, $P = 0.120$, $\eta^2 = 0.02$] or perturbation direction [$F(1,15) = 1.06$, $P = 0.319$, $\eta^2 = 0.01$], and no significant interactions among factors (all $P > 0.07$). For latency, the pattern differed. Latencies were ~6 ms longer in footplate-only conditions ($147.1 \pm 2.9$ ms) compared to whole-body conditions [$141.6 \pm 2.9$ ms; $F(1,15) = 29.89$, $P = 6.492 \times 10^{-5}$, $\eta^2 = 0.15$, see Fig. 4]. Perturbation direction also affected latency [$F(1,15) = 5.59$, $P = 0.032$, $\eta^2 = 0.05$], with toes-up trials ($145.2 \pm 2.9$ ms) slightly longer (~3 ms) than toes-down trials ($142.0 \pm 2.9$ ms).

As with amplitude, there was no main effect of motor engagement [$F(1,15) = 1.13$, $P = 0.304$, $\eta^2 = 0.01$] and no interactions were observed (all $P > 0.1$). Overall, these data show that N1 responses were robust across No-Control conditions, with amplitudes and latencies primarily influenced by whole-body sensory feedback, but not by motor engagement.

We further examined midfrontal theta activity (4–8 Hz), following the same procedure as Experiment 1 and pooling data from both directions. As before, the selected components consistently exhibited midline scalp topographies across participants, confirming that theta activity associated with postural disturbances was best captured at midline electrodes (Fig. 5*A*). Theta modulation was affected by sensory feedback and motor engagement in a manner consistent with the N1 results. Specifically, we found a significant main effect of sensory feedback on midfrontal theta power [$F(1,15) = 7.48$, $P = 0.015$, $\eta^2 = 0.107$], with larger responses during footplate-only ($2.33 \pm 0.10$ μV) compared to footplate and whole-body conditions ($2.13 \pm 0.10$ μV). In contrast, motor engagement had no significant effect on theta power [$F(1,15) = 3.05$, $P = 0.101$, $\eta^2 = 0.06$], and there was no significant interaction between motor engagement and sensory feedback [$F(1,15) = 2.97$, $P = 0.105$, $\eta^2 = 0.05$]. Overall, cortical measures (N1 amplitude

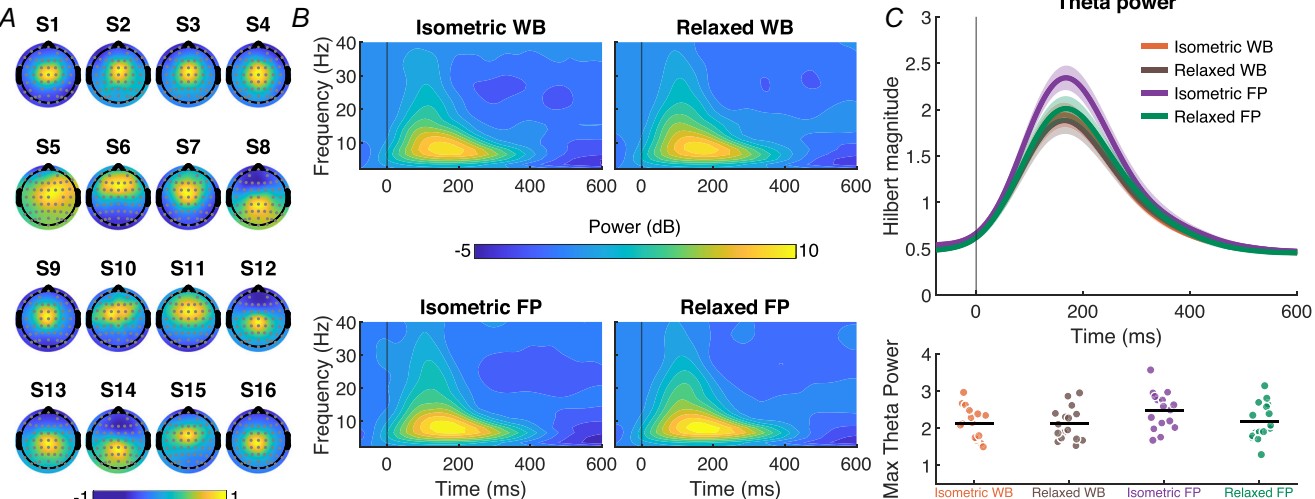

**Figure 5. Midfrontal theta modulation in Experiment 2**

*A*, participant-specific scalp topographies of the theta component identified from all trials (combining all conditions) via generalized eigendecomposition (GED), showing a consistent midline distribution across participants. Topographies were normalized within participants. *B*, time–frequency representations of power from the identified theta component, averaged across participants. Each panel corresponds to a condition combining sensory feedback and motor engagement: the left column shows conditions with isometric contractions, and the right column show trials with relaxed lower leg muscles; the top row shows responses to perturbations with combined footplate and whole-body motion (WB), while the bottom row shows responses to perturbations with footplate motion only (FP). There was a clear increase in theta-band power following perturbation onset in all conditions. *C*, to quantify these theta-band responses, we extracted the Hilbert transform magnitude of the theta component time series, band-pass filtered in the theta range (4–8 Hz). The upper panel shows the time course of theta activity averaged across participants, and the lower panel displays peak theta power values for each participant across sensory feedback and motor engagement conditions.

and theta activity) were modestly influenced by changes in sensory feedback but unaffected by motor engagement, whereas muscle activity showed the reverse pattern, depending primarily on motor engagement and not on sensory feedback. These findings extend Experiment 1 by showing that early cortical responses to perturbations persist even in conditions that are increasingly removed from active balance control, indicating that these cortical responses do not depend on the presence of motor engagement or whole-body sensory feedback during balance perturbations.

## Discussion

This study directly tested the assumption that cortical responses to balance perturbations are causally dependent on active balance control. In Experiment 1, we compared perturbation-evoked cortical and muscle responses when participants either actively controlled their balance (Balance Control) or were passively moved through the same ankle and body motion (No-Control). N1 and midfrontal theta responses were reliably evoked in both conditions and were only ∼10% smaller in No-Control conditions, whereas balance-correcting muscle activity declined by ∼30–60%. Notably, this reduction in mucles resonses indicates that the No-Control condition did not abolish all postural muscle responses, even though participants had no control authority over their motion. Nonetheless, the persistence of N1/theta despite markedly reduced corrective EMG challenges the view that these cortical signals are contingent on active balance control. To test whether cortical responses in Experiment 1 were preserved because sensory feedback and motor engagement were matched, Experiment 2 examined their persistence under No-Control conditions that were increasingly removed from balance control. When systematically removing sensory feedback (whole-body + footplate *vs.* footplate-only rotation) and motor engagement (isometric contraction *vs.* relaxed posture), we found that N1 amplitudes and midfrontal theta activity were modestly influenced only by sensory feedback, unexpectedly increasing by 8–13% during footplate-only rotations. This demonstrates that early cortical responses persist even without the sensory feedback and motor engagement of natural balance, contrasting with the implicit assumption that these cortical markers are contingent on active postural control. Instead, the persistence of these perturbation-evoked cortical signals across conditions suggests they reflect the brain's detection of unexpected sensory input.

In Experiment 1, the partial decoupling between cortical and muscular responses challenges attempts to directly link perturbation-evoked cortical markers to the corrective motor responses necessary to maintain postural control. For instance, larger N1 amplitudes in conditions requiring compensatory stepping have been taken to indicate increased cortical involvement in generating corrective actions (Payne & Ting, 2020). Our results show that cortical markers can occur independently of whether corrective actions are required, suggesting that they primarily reflect sensory detection of unexpected stimuli, making their amplitude an unreliable proxy for the degree of cortical drive to motor output. Other studies have pointed to perturbation-evoked corticomuscular connectivity as evidence for a direct cortical contribution to muscle responses, though such responses are small and inconsistent (Peterson & Ferris, 2019) or occur after rather than before muscle activity (Stokkermans et al., 2023), making them an unlikely explanation for balance corrections. An alternative case for cortical involvement has been made by Boebinger et al. (2024), who proposed that only later muscle responses (>150 ms) reflect cortical contributions and that they only emerge at increased balance demands. Using a neuromechanical model, they fit a scaled representation of cortical activity and sensory input to EMG data, suggesting a tight mapping between cortical activity and motor output. Importantly, however, cortical responses persisted during small perturbations, where cortically driven muscle activity was thought to be minimal, implying that descending cortical contributions are gated according to balance demand (Boebinger et al., 2024). Our empirical results provide more direct evidence of such gating, as perturbation-evoked cortical markers persisted in conditions without active balance demands (i.e. No-Control) that were accompanied by substantially reduced motor responses (∼30–60%) in the >150 ms time period.

This gating of motor output is also evident in other forms of postural responses, where error signals are still detected but their expression in muscle activity depends on postural context. For example, when vestibular error signals are artificially evoked using electrical stimulation, postural muscle responses are suppressed if participants are fully supported and balance corrections are unnecessary (Britton et al., 1993; Fitzpatrick et al., 1994; Luu et al., 2012). Although the vestibular-evoked sensory error signals of unexpected motion are registered by the CNS, as is evident from their conscious perception (Wardman et al., 2003), they do not translate into motor output. A similar pattern has been described for classically conditioned balance responses, where auditory cues are paired with perturbations and evoke anticipatory muscle activity even when the auditory cue is presented in isolation (Campbell et al., 2009; Kolb et al., 2004; Leeuwis et al., 2024); these conditioned responses disappear when the body is externally supported and no corrective action is required (Leeuwis et al., 2024). Together, these examples reinforce the interpretation that sensory error signals can be detected without necessarily driving post-

ural corrections. Our findings extend this principle to perturbation-evoked cortical responses, showing that they persist in the absence of balance control while associated muscle responses are suppressed.

The persistence of cortical responses in Experiment 1 may reflect continued motor engagement because participants matched torque levels similar to quiet standing; we tested this in Experiment 2 by adding a relaxed No-Control condition in which lower leg muscles were inactive. Even here, N1 amplitude and theta activity did not differ from the isometric conditions, excluding residual motor engagement as an explanation. This raises further uncertainty as to whether any changes in cortical responses reflect the need to modulate balance-correcting actions (Boebinger et al., 2024; Payne & Ting, 2020), or instead capture differences in the processing of sensory events elicited by the perturbation. Such sensory evaluation can be framed either in terms of prediction errors or as surprise signals. A prediction error reflects the brain's internal expectation of sensory consequences based on intended or ongoing motor commands, and a mismatch between predicted and actual sensory feedback produces an error signal to guide adaptations in control (Cullen, 2023; Krakauer & Mazzoni, 2011; Wolpert et al., 2011). Alternatively, a surprise response reflects the statistical improbability or salience of an event, signalling the need for heightened attention or adapted control even when no immediate correction is required (Cavanagh & Frank, 2014; Wessel et al., 2012). From either perspective, cortical responses to balance perturbations may not directly encode the motor demands of maintaining posture but rather indicate the degree to which an event violates expectations about the body's state (Adkin et al., 2006; Friston, 2005; Mierau et al., 2015; Mochizuki et al., 2010). Indeed, Payne, Ting, et al. (2019) note strong temporal, spatial and functional parallels between the balance N1 and the error-related negativity (ERN), a mid-frontal potential linked to rapid error detection in motor tasks. This may also clarify why experimental factors such as perturbation magnitude or postural stability modulate cortical responses (Payne, Hajcak, et al., 2019; Stokkermans et al., 2022): these factors scale the prediction error and/or surprise evoked by the disturbance, rather than specify the motor requirements of balance recovery. Future work should aim to disentangle and clarify how sensory prediction errors and/or surprise may relate to the neural processes that ultimately shape balance behaviour.

Other studies have proposed that these cortical signals reflect action monitoring mechanisms – neural processes that assess whether a corrective response is needed (Payne, Ting, et al., 2019; Solis-Escalante et al., 2021; Stokkermans et al., 2022). Under this framework, N1 and theta activity reflect the activation of a cortical system in the SMA, part of the posterior medial frontal cortex (pMFC) that pre-

dicts the need for corrective actions (e.g. a step) when perturbations occur. Action (or performance) monitoring is typically context-driven, with the pMFC monitoring errors signalled by subcortical structures such as the brainstem and cerebellum (for a review, see Ullsperger et al., 2014). However, in our No-Control condition, there was no context resembling active balance control, no postural goal to achieve and no opportunity for corrective actions, yet both the N1 and theta responses persisted with only modest reductions in magnitude. This suggests that if they reflect action monitoring, the monitored action is not balance control per se, but something more general such as maintaining ankle position. In this sense, the cortical signals observed during perturbations may reflect a monitoring mechanism that is expressed when unexpected events occur and can be relevant for balance but is not exclusively tied to it. Neurophysiologically, this interpretation is consistent with the multiple functions associated with the SMA: while traditionally linked to motor planning and execution (Marlin et al., 2014; Nachev et al., 2008; Stokkermans et al., 2023; Tanji, 1994), the SMA also processes somatosensory input and contributes broadly to multisensory integration (for a review, see Marasco & de Nooij, 2023). As a result, the activity we observed, particularly in No-Control conditions where no motor responses were required, may reflect multisensory input arriving at the SMA rather than the initiation of balance-correcting motor commands.

A surprising outcome of Experiment 2 was the modest but significant increase in cortical responses ($\sim$8–13%) during footplate-only perturbations, where the whole body was fixed in place. A possible explanation for this increase is that it reflects a sensory mismatch between somatosensory and vestibular inputs. In typical standing scenarios, ankle rotation is accompanied by whole-body motion, producing aligned sensory feedback from both systems. In our whole-body conditions, angular whole-body (and presumably head) accelerations preceding the balance N1 ($\pm 5.2°/s^2$ at 100 ms) were well above vestibular perception thresholds ($\sim 0.4°/s^2$; Clark & Stewart, 1970), confirming that salient vestibular activity was present. On the other hand, in footplate-only conditions, the ankles rotated while the head remained still, disrupting alignment between the expected somatosensory and vestibular feedback that normally accompanies standing movement. This effect is unlikely to reflect a prior expectation of balance as the body remained fixed throughout; instead, the larger N1 response may represent greater multisensory incongruence. Indeed, recent studies have found that mismatched multisensory information during movement modulates cortical dynamics (Alsuradi et al., 2022; Cheng & Nordin, 2025). This pattern in our data also argues against arousal or threat as a contributing factor (Adkin et al., 2008; Payne, Hajcak, et al., 2019), because participants

typically described the footplate-only condition as least engaging, yet it produced larger N1 responses than the whole-body No-Control condition. Future experiments that systematically manipulate the congruence between vestibular and somatosensory inputs could help clarify how sensory mismatches shape cortical responses.

A potential limitation of our experimental design is that Balance Control trials always preceded No-Control trials, introducing order effects that could not be disentangled from the effects of condition. Habituation of cortical responses to repeated balance perturbations is well documented, with N1 amplitudes typically declining across trial repetitions (Mierau et al., 2015; Payne, Hajcak, et al., 2019) and with observable differences as early as the second trial (Hülsdünker et al., 2015). However, the ordering conditions in our study could not be avoided, as the motion profiles recorded during Balance Control trials were required to drive participant movement in the No-Control condition. Given that No-Control trials necessarily followed Balance Control trials, any habituation across the session would be expected to disproportionately reduce responses in the No-Control condition. This means that the differences that we observed between Balance Control and No-Control may be overestimated.

Overall, our results suggest that interpreting the meaning of the 'balance N1' (and theta oscillations) requires caution. Despite its name, the N1 is robustly evoked even without active balance control, placing it within the broader class of sensory N1 responses observed across modalities, such as the auditory N1 (Budd et al., 1998) and visual N1 (Vogel & Luck, 2000), which index early cortical processing of salient or unexpected sensory events. We argue that the balance N1 and midfrontal theta are best understood as cortical markers of sensory evaluation, reflecting processes related to prediction error and surprise rather than the active control of balance.

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

## Additional information

### Data availability statement

The data that support the findings of this study and the software used to generate the figures are available in "Data and code for "Cortical responses to balance perturbations persist without active postural control"", https://doi.org/10.34894/JTBIBW.

[Correction added on 6 April 2026, after first online publication: The DOI has been corrected to read "https://doi.org/10.34894/JTBIBW."]

## Competing interests

The authors have no conflicts of interests to disclose.

## Author contributions

D.J.: conception or design of the work; acquisition, analysis or interpretation of data for the work; drafting the work or revising it critically for important intellectual content; final approval of the version to be published; agreement to be accountable for all aspects of the work. L.M.: conception or design of the work; acquisition, analysis or interpretation of data for the work; drafting the work or revising it critically for important intellectual content; final approval of the version to be published; agreement to be accountable for all aspects of the work. M.L.: conception or design of the work; drafting the work or revising it critically for important intellectual content; final approval of the version to be published; agreement to be accountable for all aspects of the work. P.A.F.: conception or design of the work; acquisition, analysis or interpretation of data for the work; drafting the work or revising it critically for important intellectual content; final approval of the version to be published; agreement to be accountable for all aspects of the work.

## Funding

This study was supported by the Dutch Research Council (NWO) Exacte en Natuurwetenschappen, VI.Vidi.203.066 (to Patrick A. Forbes).

## Keywords

balance control, balance N1, electroencephalography, midfrontal theta, prediction error, surprise

## Supporting information

Additional supporting information can be found online in the Supporting Information section at the end of the HTML view of the article. Supporting information files available:

**Peer Review History**

