## [Peer Review History · The Journal of Physiology]

Cortical responses to balance perturbations persist without active postural control

Daphne Nadine Rani Jansen, Lucas Hendrik Mensink, Matto Leeuwis, and Patrick A Forbes
DOI: 10.1113/JP290280

Corresponding author(s): Patrick Forbes (p.forbes@erasmusmc.nl)

The following individual(s) involved in review of this submission have agreed to reveal their identity: Helen Huang (Referee #1)

Review Timeline:

Submission Date:	07-Oct-2025
Editorial Decision:	01-Dec-2025
Revision Received:	18-Dec-2025
Editorial Decision:	14-Jan-2026
Revision Received:	19-Jan-2026
Accepted:	23-Jan-2026

Senior Editor: Richard Carson

Reviewing Editor: Bettina Schwab

Transaction Report:

Re: JP-RP-2025-290280 "Cortical responses to balance perturbations persist without active postural control" by Daphne Nadine Rani Jansen, Lucas Hendrik Mensink, Matto Leeuwis, and Patrick A Forbes

Dear Dr Forbes,

Thank you for submitting your manuscript to The Journal of Physiology. It has been assessed by a Reviewing Editor and by 2 expert referees and we are pleased to tell you that it is potentially acceptable for publication following satisfactory major revision.

Please address all the points raised and incorporate all requested revisions or explain in your Response to Referees why a change has not been made. We hope you will find the comments helpful and that you will be able to return your revised manuscript within 2 months. If your article is NOT for a Special Issue, you may have 9 months to revise. If you require an extension, please contact journal staff: jp@physoc.org. Please note that this letter does not constitute a guarantee for acceptance of your revised manuscript.

REVISION CHECKLIST:

We look forward to receiving your revised submission.

Yours sincerely,

Richard Carson
Senior Editor
The Journal of Physiology

REQUIRED ITEMS FOR REVISION

- Author photo and profile. First or joint first authors are asked to provide a short biography (no more than 100 words for one author or 150 words in total for joint first authors) and a portrait photograph. These should be uploaded and clearly labelled together in a Word document with the revised version of the manuscript. See Information for Authors for further details.

- Please upload separate high-quality figure files via the submission form.

- Please ensure that the Article File you upload is a Word file.

- Papers must comply with the Statistics Policy: https://jp.msubmit.net/cgi-bin/main.plex?form_type=display_requirements#statistics.

In summary:

- If $n \leq 30$, all data points must be plotted in the figure in a way that reveals their range and distribution. A bar graph with data points overlaid, a box and whisker plot or a violin plot (preferably with data points included) are acceptable formats.

- If $n > 30$, then the entire raw dataset must be made available either as supporting information, or hosted on a not-for-profit repository, e.g. FigShare, with access details provided in the manuscript.

- 'n' clearly defined (e.g. x cells from y slices in z animals) in the Methods. Authors should be mindful of pseudoreplication.

- All relevant 'n' values must be clearly stated in the main text, figures and tables.

- The most appropriate summary statistic (e.g. mean or median and standard deviation) must be used. Standard Error of the Mean (SEM) alone is not permitted.

- Exact p values must be stated. Authors must not use 'greater than' or 'less than'. Exact p values must be stated to three significant figures even when 'no statistical significance' is claimed.

- Please include an Abstract Figure file and an Abstract Figure legend. An appropriate figure legend, which should not exceed 150 words in length, should be included in the main manuscript file. The Abstract Figure is a piece of artwork designed to give readers an immediate understanding of the research and should summarise the main conclusions. If possible, the image should be easily 'readable' from left to right or top to bottom. It should show the physiological relevance

of the manuscript so readers can assess the importance and content of its findings. Abstract Figures should not merely recapitulate other figures in the manuscript. Please try to keep the diagram as simple as possible and without superfluous information that may distract from the main conclusion(s). Abstract Figures must be provided by authors no later than the revised manuscript stage and should be uploaded as a separate file during online submission labelled as File Type 'Abstract Figure'. Please also ensure that you include the figure legend in the main article file. All Abstract Figures should be created using BioRender. Authors should use The Journal's premium BioRender account to export high-resolution images. Details on how to use and access the premium account are included as part of this email.

- Please include a full title page as part of your main article (Word) file, which should contain the following: title, authors, affiliations, corresponding author name and contact details, keywords, and running title.

EDITOR COMMENTS

Reviewing Editor: Comments for Authors to ensure the paper complies with the Statistics Policy:

- please define 'SD'

- please give precise p-values also if $p < 0.001$

Comments to the Author:

Thank you for submitting your work to The Journal of Physiology. It was reviewed by two experts who are both acknowledge its important contributions to the field. Nevertheless, they raised some points that need to be addressed. In particular, I would like to stress point 3 of reviewer #1 on movement artifacts in the EEG. Please explicitly state how movement and cable sway artifacts were reduced and discuss whether the results are robust to remaining artifacts.

Senior Editor:

Comments to the Author:

Please consider including effect size estimates (and corresponding confidence intervals) in addition to the outcomes of the various null hypothesis significance tests.

REFEREE COMMENTS

Referee #1:

This study tested the assumption that cortical N1 responses to balance perturbations are dependent on active balance control. Two experiments are presented. Experiment 1 showed that cortical responses (N1 and midline theta activity) persisted even when the need for active balance control was removed, challenging the assumption. Experiment 2 systematically investigated whether the sensory feedback and motor engagement involved preceding the balance perturbation could explain the largely unaffected N1 response when active balance was not required. The results showed that cortical N1 and midline theta activity were moderately influenced by sensory feedback but not motor engagement. In contrast, muscle activity responses were influenced by motor engagement but not sensory feedback. Overall, the study suggests that the cortical responses to balance perturbations are not dependent on active balance control but rather may reflect the brain's detection of unexpected sensory input.

The manuscript is well-written with a clear motivation and narrative. The study design and methods are rigorous and presented in detail. The discussion addresses how to interpret the findings from multiple relevant perspectives. The findings provide important insights into what N1 responses to standing balance perturbations may indicate about components of human standing balance control.

Below are generally moderate/minor suggestions to improve the manuscript.

1. Adding "stick figures" of the postures and angles of the person, forceplate, and backboard at the different states (pre-perturbation, post-perturbation) may help convey the differences in conditions more quickly for more visual people. Fully defining the axes being used would also help communicate what a positive and negative rotation and angle are.

2. This study does not include ML perturbations, so the description of the ML functionality of the robotic balance simulator was distracting as it set up an expectation of ML perturbations. Consider removing it and focusing solely on explaining the AP features and function of the robotic balance simulator.

3. In Figure 1a, add the EEG box setup to the schematic and use a person wearing shorts since EMG was recorded on the legs, versus a person wearing long pants. Knowing how the EEG box was set up for a wired EEG system is helpful, as cable sway can introduce motion artifacts to EEG signals. Also consider adding a brief statement about how motion artifacts are handled, as the only EEG artifacts discussed are eye blinks, eye movements, and muscle contractions.

4. In Table 1, the EMG values for non-relevant perturbation directions could be faded, i.e. gray, like in Figure 2B.

5. Lines 490-491, the mean N1 latency values for Balance Control do not match Table 1. Text vs. Table 1 Up: 148.8 vs. 148.0. Down: 151.2 vs. 153.3. Please check the values.

6. Line 163, the reference for the robotic device does not seem correct. It is currently Rasman et al., 2024, Learning to stand with sensorimotor delays generalizes across directions and from hand to leg effectors, Communications Biology. Qiao et al., 2023, Multidirectional human-in-the-loop balance robotic system, IEEE Robotics and Automation Letters seems more appropriate. If this is indeed a mis-citation, please correct and check other references.

7. The motivation and rationale for conducting the experiments are repeated multiple times throughout the manuscript when explaining the setup, experiments, statistics, results, findings, etc. This reviewer found the repetition to be too frequent, lengthy, and somewhat unnecessary.

Referee #2:

The authors present data showing that EEG responses to balance perturbations are evoked (partly) independently of motor state and motor response. The conclusion is that these responses primarily reflect sensory input/exafference rather than sensorimotor control or motor output.

This is an important point since it will help to avoid future misinterpretation and/or over-interpretation of evoked potentials responses recorded during balance perturbations, and during sensorimotor perturbations in general.

The paper is generally very well written and presented with clear figures and analyses. I have no major concerns about methods and results. I have some minor queries on interpretation below.

Line 586 - "balance-correcting muscle activity declined by ~30-60%." This implies that the Expt 1 No-control condition failed to fully disengage balance responses. Although I think this issue is dealt with in Expt 2, it is worth explicitly stating this limitation of Expt 1 i.e. it wasn't a fully 'No-control' condition.

Line 690-92 - "amplifies the prediction error or surprise associated with the perturbation." During the footplate only condition participants were fully aware that only the footplate would move despite not knowing the timing or direction. It doesn't seem justifiable to interpret the ensuing brain responses as due to 'surprise' or 'prediction error' purely on the evidence of the EEG alone.

Fig 3 legend - '...identified from all trials...'. I assume this means that Balance control and No-control trials were combined?

Please state this for clarity if so.

Lines 463-83 and more generally - As the authors are probably aware from their Nashner 76 citation, stretch reflexes evoked by ankle rotation perturbations are actively destabilising and therefore maladaptive. For example, during a toes-up perturbation the calf muscle stretch reflex will actively send the person further backwards into the abyss, rather than stabilise them. As Nashner showed, if you repeat the same perturbation you should observe a clear attenuation of the stretch reflex (mainly the long-latency component). In his paper, the perturbations were always in the same direction. But it seems likely that such attenuation would occur even if direction and timing is unpredictable (as for the current paper; i.e. randomly toes up/down). Since a reflex in either direction is maladaptive it would make sense to suppress it. Given that these long-latency reflexes supposedly span the cerebral cortex, they will presumably impact EEG recordings. Can the authors say whether such attenuation occurred and if there is any impact on the measured brain responses. Can this shed light on the origin of the EEG response?

General point - could the N1/theta amplitude be entirely related to arousal or cognitive focus or anxiety state, independently of precise sensory or motor state? i.e. potentially little to do with processing of sensory input, but merely a response that scales with overall level of autonomic activation? I imagine the different conditions used in this experiment produced very different levels of focus/arousal, depending upon how threatened participants felt by various perturbations, or how relevant they were for balance. Please address this in the discussion and comment as to whether your data can help to confirm or refute this.

END OF COMMENTS

Dr. Patrick A. Forbes
Department of Neuroscience
Erasmus MC, University Medical Center Rotterdam
Rotterdam, The Netherlands
email: p.forbes@erasmusmc.nl

Dear Journal of Physiology Editorial Board,

Please find enclosed our revised manuscript entitled "*Cortical responses to balance perturbations persist without active postural control*", along with our point-by-point response to the reviewers.

We thank the editors and reviewers for their thoughtful and constructive feedback, which has helped us improve both the clarity and rigor of our work. Below, we summarize the key revisions made in response to the points emphasized by the reviewers and editors:

- **Statistical reporting:**

In line with the journal's Statistics Policy and the senior editor's comments, we have improved the transparency of our statistical reporting throughout the manuscript. We now explicitly define SD (standard deviation) at first use (line 234). In addition, we have added effect size estimates where these were previously missing. We now consistently report Cohen's d together with corresponding confidence intervals for all t -tests and report effect sizes for ANOVA analyses as η^2 . Exact p -values are now provided throughout, including for results previously reported as $p < .001$. Together, these changes provide a clearer representation of both the magnitude and precision of the reported effects.

- **Motion and artifacts in EEG recordings:**

In response to concerns about movement-related artifacts, we have expanded and clarified how such artifacts were mitigated. The Methods now explicitly mention the use of actively shielded electrode cables to limit motion-related interference, as well as careful routing and fixation of the EEG cable bundle to the robotic simulator, and placement of the amplifier close behind the participant to reduce cable motion (lines 342–346). In addition, we clarify the preprocessing steps used to suppress residual artifacts, including high-pass filtering and ICA-based artifact removal (lines 347–359).

In addition, we addressed all other reviewers' comments which we feel improved the overall clarity and structure of the manuscript. Specifically, we refined Figure 1 to better reflect the experimental setup, corrected minor inconsistencies, streamlined repeated motivation across sections, and sharpened interpretational statements in the Discussion. Collectively, these revisions have improved the presentation of the experimental logic, our results, and their implications.

We believe that the manuscript is significantly strengthened because of this feedback and is well suited for the Journal of Physiology and its broad audience.

Sincerely,

Dr. Patrick A. Forbes

Below we list the original reviewers' comments (black) and our responses (blue).

Reviewer 1

1. Adding "stick figures" of the postures and angles of the person, forceplate, and backboard at the different states (pre-perturbation, post-perturbation) may help convey the differences in conditions more quickly for more visual people. Fully defining the axes being used would also help communicate what a positive and negative rotation and angle are.

We thank the reviewer for this helpful suggestion. To improve the clarity of the experimental setup and more explicitly define rotation directions, we revised Figure 1 in several ways. In Figure 1A, we updated the robot schematic to depict the direction of footplate rotation, illustrating a toes-up perturbation, and adjusted the orientation of the anterior–posterior axis arrow to indicate the positive direction of rotation. In Figure 1B, we added icons to the legend to visually summarize the different experimental conditions. The Balance Control condition is represented as a balancing person with the appearance of movement that depicts active control of balance and the effect of a perturbation. No-Control trials are illustrated with two icons that separately indicate the motor engagement (i.e., isometric or relaxed) and the whole-body sensory-feedback experienced after the perturbation (i.e., a prerecorded trajectory or no motion). Motor engagement is depicted using small calf muscles in the standing figure that are either active (red - isometric) or passive (grey - relaxed). Sensory feedback of whole-body motion is depicted as a computer icon for the prerecorded trajectory (WB) or as a lock symbol for the no motion (FP). Finally, in the whole-body position traces (Figure 1C, panel 5), we now label the initial whole-body orientation (-2.5°) and indicate anterior and posterior directions with arrows. Together, these changes provide a clearer visual definition of posture, axes, and perturbation directions before and after the perturbation.

2. This study does not include ML perturbations, so the description of the ML functionality of the robotic balance simulator was distracting as it set up an expectation of ML perturbations. Consider removing it and focusing solely on explaining the AP features and function of the robotic balance simulator.

We agree that extensive detail on ML control could create an unintended expectation that ML perturbations were part of the experimental protocol. To address the reviewer's concern, we shortened and streamlined the description of ML mechanics, limiting it to what is required to understand how participants controlled their balance in that direction. Specifically, we have removed explanation of the forces and torques that are required when the harnesses moved away from the midpoint (see lines 197-203).

3. In Figure 1a, add the EEG box setup to the schematic and use a person wearing shorts since EMG was recorded on the legs, versus a person wearing long pants. Knowing how the EEG box was set up for a wired EEG system is helpful, as cable sway can introduce motion artifacts to EEG signals. Also consider adding a brief statement about how motion artifacts are handled, as the only EEG artifacts discussed are eye blinks, eye movements, and muscle contractions.

We agree with the reviewer that the suggested changes to the figure will help represent the actual conditions of our experiments. In the revised manuscript, we have updated Figure 1A to include the EEG cap, EEG/EMG amplifier, and the corresponding cable routing. We also adjusted the diagram to show a participant wearing shorts and a representation of the EMG electrodes and cables.

We also agree that clarifying how these potential motion artifact issues were mitigated is important and have made changes to the manuscript to address this issue (see lines 342-359). In summary, to minimize motion, the EEG cable was secured to the robotic simulator at the shoulder support and was attached such that the participant could freely move their head. The cable was also secured to the hip support, ensuring minimal relative motion between the cable and the participant. The cable then descended to the amplifier which was immediately behind the participant at the approximate height of the knee to further minimize any cable motion. Furthermore, our measurement system uses electrode cables that are actively shielded to reduce interference and motion artifacts (REFA amplifier, TMSi, Oldenzaal, The Netherlands). This provides further confidence that the N1 observed in our study reflects cortical activity rather than any artifacts. These details have now been included in the manuscript.

We have also provided additional information regarding the data analysis steps to show how our preprocessing pipeline is effective in suppressing movement-related artifacts. Specifically, the EEG data were high-pass filtered at 1 Hz to attenuate slow drift and motion-related contamination, and subsequently processed using ICA, a technique used in previous studies to minimize motion artifacts (Boebinger et al., 2024; Gorjan et al., 2022; Oliveira et al., 2016). These preprocessing routines are primarily designed to remove the ICA components that the algorithm identified as reflecting facial and head muscle activity or eye movement; therefore, we additionally inspected all components visually and removed any that reflected non-neural patterns, including those consistent with residual motion artifacts. A component was manually removed if the algorithm labeled it as reflecting artifactual activity and visual inspection identified clear non-neural artifacts, including motion-related activity, line noise, cardiac signals, or channel noise. This clarification has been added to the revised manuscript (line 356-359).

We further note that recent work has demonstrated that under conditions similar to ours (i.e., perturbations delivered via the feet) the balance N1 is not attributable to motion artifacts (Payne et al., 2023). Moreover, in our own data, the footplate-only perturbations, where whole-body motion and cable movement were minimal (or absent), still evoked clear N1 and theta responses. This further supports the robustness of our findings against motion-related artifacts.

4. In Table 1, the EMG values for non-relevant perturbation directions could be faded, i.e. gray, like in Figure 2B.

We agree with this suggestion and have updated Table 1 so that non-relevant EMG values are gray.

5. Lines 490-491, the mean N1 latency values for Balance Control do not match Table 1. Text vs. Table 1 Up: 148.8 vs. 148.0. Down: 151.2 vs. 153.3. Please check the values.

We thank the reviewer for identifying this discrepancy. The values in Table 1 were correct. In the revised manuscript, we have corrected the text to match the table.

6. Line 163, the reference for the robotic device does not seem correct. It is currently Rasman et al., 2024, Learning to stand with sensorimotor delays generalizes across directions and from hand to leg effectors, Communications Biology. Qiao et al., 2023, Multidirectional human-in-the-loop balance robotic system, IEEE Robotics and Automation Letters seems more appropriate. If this is indeed a mis-citation, please correct and check other references.

The robotic device described in Qiao et al. (2023) is different from the system used in the current study. The details of the robotic system used here can be found in Rasman et al. (2024). We realize

this distinction was not sufficiently clear in the original manuscript and have now added a clarifying statement (lines 169-170) explaining that our experiments were conducted using a different version of the robotic simulator from Qiao et al. (2023).

7. The motivation and rationale for conducting the experiments are repeated multiple times throughout the manuscript when explaining the setup, experiments, statistics, results, findings, etc. This reviewer found the repetition to be too frequent, lengthy, and somewhat unnecessary.

We agree with the reviewer that some of these explanations were overly repetitive. In the revised manuscript, we have streamlined the motivation and rationale throughout the Methods and Results sections by removing redundant explanations and shortening repeated arguments.

Specifically, we removed a paragraph in the Experimental set-up that summarized both experiments (line 163), as this information is already described in the dedicated sections for each experiment. In addition, the summary of the experimental setup in the Results section has been substantially shortened (lines 459-463).

Reviewer 2

1. Line 586 - 'balance-correcting muscle activity declined by ~30-60%.' This implies that the Expt 1 No-control condition failed to fully disengage balance responses. Although I think this issue is dealt with in Expt 2, it is worth explicitly stating this limitation of Expt 1 i.e. it wasn't a fully 'No-control' condition.

We thank the reviewer for this helpful clarification. We note that our aim in Experiment 1 was not to eliminate all muscle responses, but to disengage balance control, preventing participants from having any causal influence over their own movement, even though some evoked muscle activity remained. Thus, the condition functioned as a No-Control state for balance regulation, while still resembling normal stance in sensory context and muscle activity. We agree that this resemblance could allow residual EMG to persist, and as the reviewer notes this was why Experiment 2 removed both whole-body motion cues and motor engagement. To address the reviewer's suggestion, we now clarify this point and the resulting motivation for Experiment 2 more explicitly in the revised text (lines 597-600).

2. Line 690-92 - 'amplifies the prediction error or surprise associated with the perturbation.' During the footplate only condition participants were fully aware that only the footplate would move despite not knowing the timing or direction. It doesn't seem justifiable to interpret the ensuing brain responses as due to 'surprise' or 'prediction error' purely on the evidence of the EEG alone.

We agree with the reviewer that the observed effect cannot be attributed specifically to a prediction-error or surprise mechanism. In the revised manuscript, we have removed the phrase "which amplifies the prediction error or surprise associated with the perturbation" to describe the effect in a manner that does not assume a specific mechanism (line 704).

3. Fig 3 legend - '...identified from all trials...'. I assume this means that Balance control and No-control trials were combined? Please state this for clarity if so.

We confirm that the theta component was identified using data combined across Balance Control and No-Control trials. This is now explicitly stated in the revised Figure 3 caption. The same

clarification has been added to the caption of Figure 5, where theta components were likewise identified using data pooled across conditions.

4. Lines 463-83 and more generally - As the authors are probably aware from their Nashner 76 citation, stretch reflexes evoked by ankle rotation perturbations are actively destabilising and therefore maladaptive. For example, during a toes-up perturbation the calf muscle stretch reflex will actively send the person further backwards into the abyss, rather than stabilise them. As Nashner showed, if you repeat the same perturbation you should observe a clear attenuation of the stretch reflex (mainly the long-latency component). In his paper, the perturbations were always in the same direction. But it seems likely that such attenuation would occur even if direction and timing is unpredictable (as for the current paper; i.e. randomly toes up/down). Since a reflex in either direction is maladaptive it would make sense to suppress it. Given that these long-latency reflexes supposedly span the cerebral cortex, they will presumably impact EEG recordings. Can the authors say whether such attenuation occurred and if there is any impact on the measured brain responses. Can this shed light on the origin of the EEG response?

We thank the reviewer for raising this point, which prompted us to test directly whether stretch-reflex attenuation similar to that observed by Nashner (1976) occurred in our experiment. We quantified, on a trial-by-trial basis, peak EMG responses within a 0-200 ms window for the stretched muscles (mGAS and SOL for toes-up, TA for toes-down perturbations) in both Balance Control and No-Control conditions in Experiment 1. The resulting plots (Figure R1) show individual trial amplitudes and across-participant mean responses for the 20 trials in the relevant direction. mGAS and SOL responses were generally larger than those of TA, likely reflecting high baseline plantarflexor activation associated with the 2.5° forward lean at perturbation onset, which increases reflex responses (Matthews, 1986). Importantly, we observed little evidence of systematic attenuation across repetitions: reflex amplitudes were generally stable across trials in both conditions. The only notable trend was a small reduction in TA from the first trial to subsequent trials, after which responses remained relatively consistent.

We further note that the conditions used by Nashner (1976) have important differences from our study. First, as the reviewer already noted, Nashner (1976) used perturbations in a single, repeated direction whereas we alternated the perturbation direction across trials, which may have limited trial-to-trial attenuation of the stretch reflex. Second, in Nashner's study, participants were first exposed to repeated support-surface translations (AP sway perturbations) in which the stretch reflex contributed functionally to maintaining balance. Immediately following this period, participants were suddenly exposed to direct ankle rotations where the same stretch reflex was maladaptive. This abrupt shift from a helpful to a maladaptive response pattern, together with a fixed direction of the perturbation in each condition (i.e., translation and rotation), likely provided stronger conditions for rapid attenuation than in the present study design using randomized perturbation directions.

Taken together, stretch reflex attenuation was not a prominent feature in our data and is therefore unlikely to meaningfully shape the N1 amplitudes reported here. Furthermore, because our experiments were not designed to study reflex adaptation, we do not feel confident presenting these data as an exploration of this potential phenomenon.

Figure R1. Peak responses of stretch-evoked EMG across repeated perturbations in Experiment 1 for mGAS, SOL and TA during Balance Control and No-Control conditions. Each plot displays individual-trial peak EMG responses and across-participant means for the 20 trials of the direction in which the muscle was stretched: toes-up perturbations for mGAS and SOL, and toes-down perturbations for TA. Reflex amplitudes were quantified within a 0–200 ms window following perturbation onset. Across muscles and conditions, stretch reflex amplitudes remained largely stable over repeated exposures.

5. General point - could the N1/theta amplitude be entirely related to arousal or cognitive focus or anxiety state, independently of precise sensory or motor state? i.e. potentially little to do with processing of sensory input, but merely a response that scales with overall level of autonomic activation? I imagine the different conditions used in this experiment produced very different levels of focus/arousal, depending upon how threatened participants felt by various perturbations, or how relevant they were for balance. Please address this in the discussion and comment as to whether your data can help to confirm or refute this.

We appreciate the reviewer's suggestion that arousal or anxiety may contribute to N1/theta amplitude. Indeed, prior work has shown that balance-related cortical signals can be modulated by factors associated with threat, attention, and task demands, which could plausibly covary with

autonomic activation (Adkin et al., 2008; Quant et al., 2004; Stokkermans et al., 2022; Weerdesteyn et al., 2008). However, our study provides evidence that arousal is not a major contributor to the observed effects. First, based on participants' anecdotal reports, the No-Control footplate-only trials were described as less engaging compared to No-Control whole-body trials. If N1/theta primarily reflected global arousal or threat, we would expect smaller responses in the footplate-only condition. Instead, within No-Control trials, footplate-only perturbations produced larger N1 responses. Second, participants were explicitly informed that in all No-Control conditions there was no risk of falling and that they could not influence their motion, which likely reduced perceived threat and arousal. Indeed, as shown in Experiment 1, balance correcting muscle responses decreased substantially in the No-Control trials, while the N1 and theta responses remained robust and differed only modestly from Balance Control.

Taken together, these observations argue against threat or arousal as the primary driver of the cortical differences we report and instead support an interpretation based on the sensory characteristics of the perturbation. We have added a statement in the Discussion to clarify this point (lines 706-709), noting that our findings do not support differences in threat or arousal as the main explanation for the modulation of N1/theta amplitude. Future studies that include concurrent autonomic recordings could further confirm our interpretation.

References

- Adkin, A. L., Campbell, A. D., Chua, R., & Carpenter, M. G. (2008). The influence of postural threat on the cortical response to unpredictable and predictable postural perturbations. *Neurosci Lett*, 435(2), 120-125. <https://doi.org/10.1016/j.neulet.2008.02.018>
- Boebinger, S., Payne, A., Martino, G., Kerr, K., Mirdamadi, J., McKay, J. L., Borich, M., & Ting, L. (2024). Precise cortical contributions to sensorimotor feedback control during reactive balance. *PLOS Computational Biology*, 20(4), e1011562. <https://doi.org/10.1371/journal.pcbi.1011562>
- Gorjan, D., Gramann, K., De Pauw, K., & Marusic, U. (2022). Removal of movement-induced EEG artifacts: current state of the art and guidelines. *J Neural Eng*, 19(1). <https://doi.org/10.1088/1741-2552/ac542c>
- Matthews, P. B. (1986). Observations on the automatic compensation of reflex gain on varying the pre-existing level of motor discharge in man. *J Physiol*, 374, 73-90. <https://doi.org/10.1113/jphysiol.1986.sp016066>
- Nashner, L. M. (1976). Adapting reflexes controlling the human posture. *Experimental Brain Research*, 26(1), 59-72. <https://doi.org/10.1007/BF00235249>
- Oliveira, A. S., Schlink, B. R., Hairston, W. D., König, P., & Ferris, D. P. (2016). Induction and separation of motion artifacts in EEG data using a mobile phantom head device. *J Neural Eng*, 13(3), 036014. <https://doi.org/10.1088/1741-2560/13/3/036014>
- Payne, A. M., Ting, L. H., & Hajcak, G. (2023). The balance N1 and the ERN correlate in amplitude across individuals in small samples of younger and older adults. *Experimental Brain Research*, 241(10), 2419-2431. <https://doi.org/10.1007/s00221-023-06692-9>
- Qiao, C. Z., Nasrabadi, A. M., Partovi, R., Belzner, P., Kuo, C., Wu, L. C., & Blouin, J.-S. (2023). Multidirectional Human-in-the-Loop Balance Robotic System. *IEEE Robotics and Automation Letters*.
- Quant, S., Adkin, A. L., Staines, W. R., & McIlroy, W. E. (2004). Cortical activation following a balance disturbance. *Experimental Brain Research*, 155(3), 393-400. <https://doi.org/10.1007/s00221-003-1744-6>
- Rasman, B. G., Blouin, J.-S., Nasrabadi, A. M., van Woerkom, R., Frens, M. A., & Forbes, P. A. (2024). Learning to stand with sensorimotor delays generalizes across directions and from hand to leg effectors. *Communications Biology*, 7(1), 384. <https://doi.org/10.1038/s42003-024-06029-4>
- Stokkermans, M., Solis-Escalante, T., Cohen, M. X., & Weerdesteyn, V. (2022). Midfrontal theta dynamics index the monitoring of postural stability. *Cereb Cortex*. <https://doi.org/10.1093/cercor/bhac283>
- Weerdesteyn, V., Laing, A. C., & Robinovitch, S. N. (2008). Automated postural responses are modified in a functional manner by instruction. *Exp Brain Res*, 186(4), 571-580. <https://doi.org/10.1007/s00221-007-1260-1>

Dear Dr Forbes,

Re: JP-RP-2025-290280R1 "Cortical responses to balance perturbations persist without active postural control" by Daphne Nadine Rani Jansen, Lucas Hendrik Mensink, Matto Leeuwis, and Patrick A Forbes

Thank you for submitting your revised Research Article to The Journal of Physiology. It has been assessed by the original Reviewing Editor and Referees and has been well received. Some final revisions have been requested.

REVISION CHECKLIST:

We look forward to receiving your revised submission.

Yours sincerely,

Richard Carson
Senior Editor
The Journal of Physiology

REQUIRED ITEMS FOR REVISION

- Papers must comply with the Statistics Policy: https://jp.msubmit.net/cgi-bin/main.plex?form_type=display_requirements#statistics.

In summary:

- If n {less than or equal to} 30, all data points must be plotted in the figure in a way that reveals their range and distribution. A bar graph with data points overlaid, a box and whisker plot or a violin plot (preferably with data points included) are acceptable formats.
 - If $n > 30$, then the entire raw dataset must be made available either as supporting information, or hosted on a not-for-profit repository, e.g. FigShare, with access details provided in the manuscript.
 - 'n' clearly defined (e.g. x cells from y slices in z animals) in the Methods. Authors should be mindful of pseudoreplication.
 - All relevant 'n' values must be clearly stated in the main text, figures and tables.
 - The most appropriate summary statistic (e.g. mean or median and standard deviation) must be used. Standard Error of the Mean (SEM) alone is not permitted.
 - Exact p values must be stated. Authors must not use 'greater than' or 'less than'. Exact p values must be stated to three significant figures even when 'no statistical significance' is claimed.
 - Please include an Abstract Figure file and an Abstract Figure legend. IMPORTANT - An appropriate figure legend, which should not exceed 150 words in length, should be included in the main manuscript file.
- Thank you for submitting your revised Research Article to The Journal of Physiology. It has been assessed by the original Reviewing Editor and Referees and has been well received. Some final revisions have been requested.

EDITOR COMMENTS

Reviewing Editor:

Thank you for submitting the revision of your manuscript to The Journal of Physiology. Apart from two small comments of referee #1 that still need to be addressed, all concerns have been addressed. I congratulate the authors on their important work.

REFeree COMMENTS

Referee #1:

Thanks to the authors for the improved manuscript and detailed response. The authors have addressed my comments well. One thing to clarify in Fig. 1 is that the amplifier is for both EEG and EMG, not just EMG, as it is currently labeled EMG. Also,

there is a typo in the Fig. 1 caption when describing the added icons - "foorplate + whole-body". Thanks for the nice study.

Referee #2:

I am happy that the authors have addressed my concerns. The analysis of trial-by-trial changes in response amplitude is particularly welcome, and shows there is minimal habituation/adaptation.

END OF COMMENTS

Dr. Patrick A. Forbes
Department of Neuroscience
Erasmus MC, University Medical Center Rotterdam
Rotterdam, The Netherlands
email: p.forbes@erasmusmc.nl

Dear Journal of Physiology Editorial Board,

Please find enclosed our revised manuscript entitled "*Cortical responses to balance perturbations persist without active postural control*".

We thank the editors and reviewers for the positive feedback on our manuscript and the opportunity to submit these final revisions. Below are the point-by-point responses to the reviewers. We further note that we have made some esthetic changes to the Graphical Abstract to improve the representation of our findings, primarily adding key terms to allow a less-informed viewer to understand what the study is addressing.

We believe that the manuscript is significantly strengthened through the constructive process of these reviews.

Sincerely,

Dr. Patrick A. Forbes

Below we list the original reviewers' comments (black) and our responses (blue).

Reviewer 1

1. Thanks to the authors for the improved manuscript and detailed response. The authors have addressed my comments well. One thing to clarify in Fig. 1 is that the amplifier is for both EEG and EMG, not just EMG, as it is currently labeled EMG. Also, there is a typo in the Fig. 1 caption when describing the added icons - "foorplate + whole-body". Thanks for the nice study.

We thank the reviewer for these helpful notes regarding clarity and typos. We have modified Figure 1 to reflect the measurement of EEG on the amplifier. We have also corrected the typo in the caption.

Reviewer 2

1. I am happy that the authors have addressed my concerns. The analysis of trial-by-trial changes in response amplitude is particularly welcome, and shows there is minimal habituation/adaptation..

We thank the reviewer for motivating us to address these key issues and strengthen our manuscript.

Dear Dr Forbes,

Re: JP-RP-2026-290280R2 "Cortical responses to balance perturbations persist without active postural control" by Daphne Nadine Rani Jansen, Lucas Hendrik Mensink, Matto Leeuwis, and Patrick A Forbes

We are pleased to tell you that your paper has been accepted for publication in The Journal of Physiology.

Yours sincerely,

Richard Carson
Senior Editor
The Journal of Physiology

IMPORTANT POINTS TO NOTE FOLLOWING ACCEPTANCE OF YOUR PAPER:

- **IMPORTANT NOTICE ABOUT OPEN ACCESS:** To assist authors whose funding agencies mandate immediate public access to published research findings, The Journal of Physiology allows authors to pay an Open Access (OA) fee to have their papers made freely available immediately on publication.

- You can help your research get the attention it deserves! Check out Wiley's free Promotion Guide for best-practice recommendations for promoting your work at: www.wileyauthors.com/eeo/guide. You can learn more about Wiley Editing Services which offers professional video, design, and writing services to create shareable video abstracts, infographics, conference posters, lay summaries, and research news stories for your research at: www.wileyauthors.com/eeo/promotion.

- If you would like to receive our 'Research Roundup', a monthly newsletter highlighting the cutting-edge research published in The Physiological Society's family of journals (The Journal of Physiology, Experimental Physiology, Physiological Reports, The Journal of Nutritional Physiology and The Journal of Precision Medicine: Health and Disease), please click this link, fill in your name and email address and select 'Research Roundup':
<https://www.physoc.org/journals-and-media/membernews>

EDITOR COMMENTS

All final comments have been addressed. I congratulate the authors on their nice work.